# Status of Plant Protein-Based Green Scaffolds for Regenerative Medicine Applications

**DOI:** 10.3390/biom9100619

**Published:** 2019-10-17

**Authors:** Hossein Jahangirian, Susan Azizi, Roshanak Rafiee-Moghaddam, Bahram Baratvand, Thomas J. Webster

**Affiliations:** 1Department of Chemical Engineering, College of Engineering, Northeastern University, 360 Huntington Avenue, Boston, MA 02115, USA; roshanak.rafiee@gmail.com; 2Applied Science and Technology Education Center of Ahvaz Municipality, Ahvaz 617664343, Iran; azisusan@gmail.com; 3Department of Physiotherapy, Faculty of Health and Sport, Mahsa University, Bandar Saujana Putra, Jenjarum Selangor 42610, Malaysia; b.baratvand@gmail.com

**Keywords:** biomaterials, green scaffolds, plant proteins, tissue engineering

## Abstract

In recent decades, regenerative medicine has merited substantial attention from scientific and research communities. One of the essential requirements for this new strategy in medicine is the production of biocompatible and biodegradable scaffolds with desirable geometric structures and mechanical properties. Despite such promise, it appears that regenerative medicine is the last field to embrace green, or environmentally-friendly, processes, as many traditional tissue engineering materials employ toxic solvents and polymers that are clearly not environmentally friendly. Scaffolds fabricated from plant proteins (for example, zein, soy protein, and wheat gluten), possess proper mechanical properties, remarkable biocompatibility and aqueous stability which make them appropriate green biomaterials for regenerative medicine applications. The use of plant-derived proteins in regenerative medicine has been especially inspired by green medicine, which is the use of environmentally friendly materials in medicine. In the current review paper, the literature is reviewed and summarized for the applicability of plant proteins as biopolymer materials for several green regenerative medicine and tissue engineering applications.

## 1. Introduction

Tissue engineering (a subset of regenerative medicine) is a multidisciplinary field that includes the ”use of the principles and techniques of engineering and biological knowledge towards the essential comprehension of structure-function connections in pathological and normal mammalian tissues and the growth of biological alternatives which regenerate, preserve or recover tissue function” [1].

One of the principle approaches in tissue regeneration includes the in vitro growth of related cells into three-dimensional (3D) scaffolds. 3D micro-environments are provided by porous scaffolds that can simulate the natural extracellular matrix (ECM) and can permit cell penetration and extensive areas for matrix embedding via cells to regenerate new tissue. A cell-seeded scaffold is either cultivated in vitro to generate new tissue that can subsequently be transplanted into a damaged part, or is directly implanted within the damaged part, by the body’s own systems, wherever restoration of an organ or tissue is stimulated in vivo (Figure 1). Hence, the scaffold is an essential element for tissue engineering. A perfect scaffold should inspire the generation of tissue that is physically and functionally strong, while being biocompatible and economical to produce [2,3]. Through such stringent requirements, many research groups have employed toxic solvents and/or have developed polymers that are not natural to our environment. They do not represent green technologies which numerous other fields have embraced to save our planet. One boundary of tissue engineering lies in the employment of green scaffolds for cell-based remedies. This is because since the goal of tissue engineering is to formulate a natural tissue, many have speculated that green materials should first and foremost be used.

Among natural polymers, proteins may be the most valued and under-used polymeric starting materials for the growth of novel products for biotechnological and biomedical applications [4,5,6]. The utilization of natural proteins to produce biomaterial-based scaffolds is an interesting idea due to their intrinsic bioactivity, biocompatibility, degradation property, and natural binding segments, which can be suitable to control cell attachment and tissue development both in vitro and in vivo [7].

Although not commonplace, in the last decade, many protein and protein-based composites have been studied by green technologists for modern biomedical applications [8,9]. Animal proteins (such as albumin, gelatin, and collagen) and plant proteins (such as zein, soy protein, and wheat gluten) have been generally explored for such medical applications. Plant proteins are plentiful and economical which can be fabricated into hydrogels, micro/nanoparticles, fibers, and porous structures with properties suitable for different medical applications (such as tissue engineering, drug delivery, etc.). Most importantly, these proteins have inherent water stability because they contain abundant intra-and intermolecular disulfide bonds and are environmentally-friendly. Importantly, plant-derived proteins have less immunogenicity potential and certainly are less likely for disease transmission compared to animal proteins [10,11,12]. Additionally, they possess a comparatively low molecular weight in comparison to animal proteins and display greater polarity; thus, they are naturally hydrophilic and potentially efficacious for cell adhesion.

Because of the merits of plant proteins, increasing attempts are being made to probe possibilities to develop such biomaterials (including micro and nanofibers, hydrogels, porous, and micro and nanoparticles) from numerous plant derived proteins for tissue engineering applications [13,14], drug delivery systems, [15,16] and wound dressings [17,18,19]. However, plant proteins (particularly soy proteins and gluten) possess confined organic solvents (albeit much less toxic than traditional polymeric tissue engineering solvents such as chloroform), and thus, it is hard to produce biomaterials from such proteins. Furthermore, plant protein-based biomaterials mostly possess insignificant physiochemical and biological properties. Blending with synthetic polymers and crosslinking are some of the typical methods that can be applied to enhance their mechanical properties, water resistance, biocompatibility, and biological behaviors and make them appropriate for medical applications. These approaches also reduce the use of non-environmentally friendly, traditional tissue engineering polymers. Figure 2 illustrates the several advantages of using plant proteins as scaffolds for tissue engineering, which includes reasons for their needed high attention in medical applications. Some of the properties of plant proteins which are suitable for medical applications have been compared to the properties of silk and collagen in Table 1.

This review will provide a summary of recent research for using plant protein-based biomaterials in different forms such as porous, hydrogels, and micro- and nano-fibers, as well as microsphere scaffolds for numerous tissue engineering applications.

## 2. Various Forms of Scaffolds for Tissue Engineering Applications

Scaffolds play an essential role in the restoration and more significantly regeneration of tissues by providing for a proper stage for tissue growth, potentially allowing for a significant supply of different factors related to cell viability, proliferation and differentiation of cells [37]. Scaffolds can be classified into different forms, such as porous, fibrous, hydrogels, microspheres, composites, and acellular scaffolds (Figure 3). All of these types of scaffolds have advantages and also significant drawbacks, not to mention being unfriendly to the environment (Table 2).

### 2.1. Porous Scaffolds

Various forms of porous scaffolds exist, such as foams, meshes, sponges [39], and micro- and nanoscale fibers; the last two forms can be classified under fibrous scaffolds [40]. Porous scaffolds are mostly fabricated by: (i) utilizing porogens to control the preferred shape and the pore size in the scaffold; (ii) prototyping; (iii) layer-by-layer cell and woven or non-woven nanoscale fibers through the electrospinning method; and (iv) the most latest, 3D or even four-dimensional (4D) printing [41,42]. Porous scaffolds have been applied for developing tissues, organs, and rigid tissues (for example bone) [43,44,45,46,47,48,49,50,51,52,53,54].

### 2.2. Fibrous Scaffolds

As cited earlier, fibrous scaffolds can indeed be classified as porous scaffolds and can be fabricated from nano-scale fibers and possess a high potential to simulate the normal environment of human tissue. Interestingly, although those that promote the use of nanofibers stipulate that they mimic the natural tissue more than micron fibers, such researchers still employ environmentally unfriendly polymer chemistries or processes. Nano-scale fibers are made by methods such as phase separation, self-assembly, drawing, melt blowing, template synthesis, centrifugal spinning, and the more extensively applied electrospinning approaches [55,56]. The nano-scale fibers are sometimes precisely functionalized via a simple blending (or mixing) or coating procedure, or through surface grafting polymerization for embedding ligand molecules and adhesive proteins onto the nanofiber surface [47,48,49,50,52]. Nanofibrous scaffolds are broadly used for rigid and soft tissue engineering applications.

### 2.3. Scaffolds Based on Hydrogels

Hydrogel scaffolds, which are prepared from natural or synthetic polymers [57,58,59,60], have a high potential for use in biomedical fields because of their biodegradable and biocompatible characteristics as well as their capability to promote inherent cellular interactions [38]. It is important to mention that just because a polymer is biodegradable, that does not make it environmentally friendly, as the degradation products can still be quite toxic to the environment. The latest advancements in the design and use of hydrogels has led to considerable developments in tissue engineering and drug delivery [61]. To guide the development of new tissues, those hydrogels which are composed of mostly synthetic polymers possess many advantages over those hydrogels comprised of natural polymers [38]. Cell encapsulated hydrogels (for injectable use in tissues) can be formed by mixing the monomeric solution with the cells [62,63]. Such hydrogels are formed by covalent or ionic crosslinking of diverse polymers that are biocompatible to easily encapsulate living cells or drug compounds. In a study, a microsphere hydrogel was fabricated by 1% collagen microspheres into a 0.3% collagen bulk for dermal regeneration [64]. The elastic properties and swelling ability of hydrogels makes them suitable for injectable purposes and bio-printing uses, which is a developing technology for the 3D production of structures utilized for the fabrication of complex functional tissues and artificial organs, from nano- to macro-scales. More recently, 3D printing techniques have been used for the fabrication of hydride hydrogel scaffolds with potential for tissue repair [65,66,67]. For example, Giustina et al. reported the 3D fabrication of methacrylated pullulan structures with feature dimensions ranging from millimeters to microns by using two light (visible stereolithography and two photon lithography) assisted 3D printing technologies for tissue engineering applications [65]. In another study, bacterial cellulose nanofibers (BCNFs) were applied to improve the structural resolution and mechanical properties of 3D printed scaffolds composed of silk fibroin and gelatin for soft tissue regeneration [66]. Zhang et al. prepared chitosan/silk composite scaffolds using silk particles, and silk microfibers and nanofibers via a 3D printing method for soft tissue engineering applications. They found that nanofibers reinforced scaffolds offering the greatest increase in stiffness, cell attachment and best printing accuracy [67]. Thus, it is clear that nanofibers can outperform micron fibers in tissue engineering, but one must also concentrate on using environmentally friendly materials.

### 2.4. Microsphere Scaffolds

In recent years, scaffolding procedures using microspheres have merited substantial attention [68]. In modern tissue engineering, such scaffolds are mainly employed for drug delivery or for gene therapy [69]. Microsphere scaffolds can be formed through different methods such as heat sintering [70], solvent/non-solvent sintering [71], or sintering only with a non-solvent method [68], and solvent vapor treating [72]. These methods offer some advantages, for example simple production, controlled morphological structure and physicochemical behaviors, resulting in multipurpose materials for the pharmacokinetics of the captured molecules [73]. It is also important to emphasize work that uses solvent-free sintering to avoid the use of solvents harming our environment [68].

### 2.5. Polymer/Bioceramic Composite Scaffolds

The utilization of polymer/bioceramic composites may be advantageous for some tissue engineering uses, as it is valuable in terms of organizing the behaviors of the materials in order to promote promising physiological replies from the host tissue [74]. There are primarily three types of ceramics applied in regenerative medicine scaffold production: (i) non-absorbable, that are intrinsically inert, such as zirconia, alumina, nitrides, carbons, and silicone; (ii) semi-inert surface reactive, for example dense hydroxyapatite (HAP) (bio-reactive) and glass ceramics; and (iii) biodegradable ceramics, which are non-inert in nature, e.g., HAP, aluminum calcium phosphate, tricalcium phosphate (TCP), coralline, and plaster of Paris [75]. The use of ceramics possesses some advantages, for instance, resistance to oxidization, compatibility, and great compression, though brittleness, difficult production, high density, and deficiency of reliability and flexibility are several major difficulties of ceramics. Polymers have low tensile strength and modulus, but are flexible, while ceramics are stiff. Thus, polymer composites prepared from polymers and bioceramics have enhanced mechanical properties as well good degradability behavior [76]. More recently, several research groups have reported fabrication of bioceramic composite scaffolds based on natural/synthetic polymers in combination with nano-hydroxyapatite (n-HAP) [77,78,79,80,81], or a combination of bioceramic particles [50] for bone tissue regenerations. Moreover, while HAP has been a staple of attention for bone tissue engineering, highly crystalline HAP can be non resorable and, thus, present an environmental hazard. Much more effort needs to be paid to the design and use of environmentally friendly ceramics for regenerative medicine.

### 2.6. Acellular Scaffolds

Acellular tissue matrices are produced via the removal of cellular components from tissues to create collagen-rich matrices that assist in the growth of cells and regeneration of tissues. Acellular tissue scaffolds are made through decellularization approaches based on physical, chemical, or enzymatic decomposition. Such approaches include trypsin/(ethylene diamine tetra-acetic acid) EDTA treatment, hypertonic or hyptonic solution treatment, repeated freeze-thaw cycles, etc. This permits the texture and biochemical structure of the decellularized material to be retained as approximately close to its original form as possible, so that the material can be applied as an efficacious substitute to regenerate the injured tissue [41,82]. Of course, environmentally friendly chemicals and approaches must be embraced as even trypsin/EDTA (which is used to lift cells from underlying matrices) presents environmental hazards. Such scaffolds have revealed superior regenerative influences in genitourinary tissues without immunogenic rejection [83,84]. Acellular tissue scaffolds have been used to regenerate human tissues like for the regeneration of the intestines, heart, breast, esophagus, spine, dura mater, or urinary bladder [85,86,87,88,89,90,91].

Recent cellular and acellular studies using various forms of scaffolds for tissue engineering applications are listed in Table 3.

## 3. Plant Proteins

### 3.1. Chemical and Physical Treatment

In general, plant proteins possess lower molecular weights as compared with animal proteins and, therefore, are predisposed to enzymatic decomposition in the mammalian body. A series of bioactive and physical properties can be obtained with protein scaffolds via chemical and physical treatments. Chemical treatments, for instance, crosslinking, are, as a result, important for obtaining the required mechanical properties and aqueous stability of plant protein-based scaffolds throughout the implantation time. Aldehydes, such as formaldehyde and glutaraldehyde, although possessing environmental risks, have been extensively applied as crosslinking agents to obtain improved protein characteristics [95,96]. Formaldehyde is generally used as a crosslinking agent, and it can interact with the amino acids of the protein chains for example tyrosine, tryptophan, arginine, histidine, and cysteine. Glutaraldehyde is more particular as compared to formaldehyde and it interacts with cysteine, histidine, tyrosine, and lysine. The water stability of zein fibers was enhanced through crosslinking with glutaraldehyde [97]. Hernandez-Munoz et al. [98] studied the physical properties of glutenin films treated with aldehyde. They stated that the barrier property of gluten samples decreased by about 30% when glutaraldehyde, formaldehyde, and glyoxal were used. It has also been found that formaldehyde offers the largest tensile strength enhancement followed by glutaraldehyde and glyoxal. However, the toxicity of aldehydes is a major drawback that must be considered when producing biodegradable materials. Owing to the toxicity of the aldehyde, many studies have been carried out to enhance plant protein-based scaffolds properties by using natural crosslinking agents. The effects of gallic acid and tannins as natural crosslinking agents on the properties of sunflower protein isolate films were studied as an environmentally friendly solution. Results showed that the addition of tannins and gallic acid in films offer greater tensile strength than for control samples [99]. The largest tensile strength improvement obtained for those films were prepared with Chestnut and Tara tannin. The tensile strength increased by 50% and ~57% for 3.5% and 6.0% of tannin, respectively; however, it was less than those films made by using aldehyde. The incorporation of 1.50% glutaraldehyde improved the tensile strength of the film by ~86% without a decline in elongation at break [99]. This may be explained by weak connections created by tannin when compared to those strong covalent bonds formed by aldehyde.

Transglutaminase, a microbial enzyme, was added to soy proteins to modify their mechanical properties and texture of films [100,101,102]. The treatment by four units per soy protein isolate (SPI) (Ug(-1)) of microbial transglutaminase enhanced the tensile strength and surface hydrophobicity by 10–20% and 17–56%, respectively [102]. An earlier study described that transglutaminase can improve the mechanical properties of soy protein scaffolds by creating intramolecular or intermolecular covalent bonds [100,101,102]. Transglutaminase effectively improved the gel strength, water holding capacity, and viscoelasticity of high-fiber tofu [103].

Irradiation is a physical treatment that is applied to induce modification in protein chains [104]. It has been found to be an efficient technique to enhance barrier and mechanical properties of protein films. Irradiation affects protein chains either through amino acid oxidation, alterations in amino acid conformation, fragmentation, the production of protein free radicals, or via an increase in covalent bonds [105]. The water vapor permeability of the soy protein films decreased 13%, and its tensile strength increased two times by γ-irradiation [106]. γ-irradiation up to 10 kGy led to higher elongation at break in cellulose/soy protein isolate films [107]. γ-irradiation on gluten films increased its tensile strength by ~49% and reduced water vapor permeability by ~29% in comparison with non-treated films [108]. Ultraviolet (UV) radiation causes the modification of proteins which has been recognized to form cross-links and principally involves aromatic amino acid residues [109].

Thermal heating is a physical treatment and generally happens above a specific threshold temperature. The heat treatment of a protein film leads to increased crosslinking of the hydrogen, disulphid, and hydrophobic bonds, thus, creating new structural arrangements [110]. Hence, plant protein based films treated under an augmented temperature display considerably enhanced tensile strength [111,112,113]. The heat treatment of proteins induces thermoplasticity, which permits a broad variety of structures and shapes to be produced, consisting of granules/pellets, films, and gels [102,114]. Microwave and ultrasonic treatment possess the potential to modify the properties of soy proteins [115].

Hydrostatic pressure is a physical treatment which induces gel formation in proteins. Hydrostatic pressure disturbs electrostatic interactions and intermolecular hydrophobic interactions, augments the reaction between sulfhydryl groups and stabilizes hydrogen bonding in proteins. Under pressure, proteins unfold and, if their concentration is highly sufficient, they make gel nets and sediment [116]. Soy protein gels with high water holding capacity have been formed under high pressure [117]. Hydrostatic pressure also does not often involve the use of toxic solvents, representing a green manufacturing process.

Another procedure is to blend proteins with other synthetic and natural polymers to produce new materials with enhanced performance than their individual components. Thus far, plant proteins have been combined with a number of biodegradable polymers and plasticizers, for example, proteins, polysaccharides, and synthetic polymers, in order to obtain desirable properties. The gelatin/zein nanofibers can cross-link using glucose. Such cross-linked fiber mats presented good long-term water resistance, and tunable wettability and mechanical properties, and significant biocompatibility without cytotoxicity [118]. Furthermore, polyethylene oxide (PEO) was mixed with zein in 80% ethanol aqueous solutions to prepare electrospun fiber mats suitable for use as a tissue engineering scaffold, wound dressing and for food packaging [119]. In another study, chitosan was blended with zein to prepare blended films via casting. The prepared films showed enhanced mechanical and barrier properties [120]. Zein films were modified by a two-step process consisting of combining with chitosan, followed by exposure to cold plasma [121]. Cold plasma is an eco-friendly and chemical free approach which seems to be a promising method for zein based films in today’s industries. Chitosan oligosaccharide (COS) was grafted on hydrolyzed wheat gliadin (HWG) using microbial transglutaminase (MTGase) as a catalyst [122].

Water is the best effectual plasticizer for proteins and, of course, the most environmentally friendly [36]. Glycerol [123,124], polyethylene glycol (PEG) [125], and sorbitol [126,127] are generally used as plasticizers for protein films; however, some use toxic solvents during preparation. Zein films plasticized with oleic acid (OA) presented higher water resistance, UV-light opacity, and water barrier properties than glycerol-plasticized films [128]. Hong et al. [129] used coconut oil as a plasticizer in soy protein isolates blended with poly caprolactone (PCL). The results exhibited that ~20 mL of plasticizer was sufficient to plasticize SPI in the PCL: SPI blend systems comprising of 10, 20, and 30% SPI and such systems were confirmed through crystallinity and melting point modification. The different treatment approaches are illustrated in Figure 4.

### 3.2. Plantprotein-Based Nanocomposites

The combination of nano-sized fillers with a polymer matrix resulting in a single structure with significantly enhanced mechanical and biological properties also seems to be a strategy for better tissue engineering. Special attention has been ascribed to nanocomposites for bone tissue engineering and regeneration owing to the nano-sized structures of the fillers which can significantly increase the tissue bonding capacity of the polymeric matrices that singular materials cannot achieve hence permitting the production of better biomaterials. Two categories of nano-sized fillers are generally used, inorganic fillers such as metal-based nanoparticles, carbon nanotubes and layer silicates, and organic fillers like nanofibrillar cellulose, polysaccharide whiskers, carbon black, etc. There are only a few reports on the use of nano-sized fillers into plant proteins as tissue engineered scaffolds, and they were fabricated as composites with nano-sized fillers mostly for food packaging.

Nano-hydroxyapatite (n-HAP) is a favorable reinforcing nanomaterial to fabricate composite scaffolds for bone tissue regeneration. However, as discussed before, the highly crystalline structure of HAP can be an environmental hazard. In a recent study, the tensile strength and modulus of soy protein isolate (SPI) sheets were improved by the uniform dispersion of n-HAP synthesized from eggshell waste in an SPI matrix [130]. Important enhancements in tensile strength, Young’s modulus, and thermal stability of SPI nanocomposite films were obtained through the combination of well-dispersed eggshell nanopowder (ESNP), owing to its high crystallinity, surface area, stiffness, and thermal degradation temperature of the nano-sized fillers [131]. Incorporation of SiO_2_ improved the thermal stability and tensile strength and decreased the barrier properties of the SPI films [132]. In another study, electrospun zein nanofibers were coated with calcium phosphate nanosheets suitable for bone tissue regeneration. The mineralized electrospun zein scaffolds improved specific biological functions such as adhesion, spreading and proliferation of adipose-derived stem cells resulting from the reserved fibrous morphology and the bioactive environment provided via calcium phosphate minerals [133].

Clay is generally used as a filler to reinforce a wide range of polymers. Nanoclay (MMT-Na^+^) enhanced the mechanical properties and water uptake capacity as well rheological properties of SPI when its content was above the percolation threshold concentration in a SPI/(MMT-Na^+^) nanocomposite [134]. In a similar study, it was shown that the mechanical and water vapor barrier properties of zein/nanoclay nanocomposites increased in the presence of small amounts of MMT (up to 3%) [135].

Conductive biofoams have been made from glycerol-plasticized wheat gluten (WGG) with carbon nanotubes (CNTs), carbon black (CB) or reduced graphene oxide (rGO) as the conductive filler phase by conventional freeze-drying. The CNT-filled foams exhibited conductivity higher than foams filled with the CB particles, and the rGO-filled foams offered a conductivity lower to that achieved with the CNTs or CB particles, which is described as being associated to the sheet-like morphology of the rGO flakes [136].

The incorporation of cellulose nanofibril (CNF) into soy protein/*Cedrus deodara* pine needle extract (PNE)/lactic acid composites greatly enhanced the tensile strength of composite films due to the filling effects of CNF. Furthermore, CNF modified the antimicrobial performance of composite films by decreasing the release of PNE and lactic acid from the film matrix [137]. A recent study has shown that the incorporation of CNF into wheat gluten/carboxymethyl cellulose matrix could enhance the mechanical properties and decrease barrier properties of films due to a reduction of pores and cavities in the nanocomposite structure [138].

By combining certain antibacterial nanoparticles, such as silver, titanium dioxide (TiO_2_), zinc oxide (ZnO), etc., plant proteins can present antibacterial, antiviral, and antioxidant activities, which make them suitable for use in biomedical fields. However, synthesizing such nanoparticles may use chemical and toxic materials hazardous for the environment. Thus, researchers have attempted to find green raw materials and processes which have potential to synthesize metal-based nanoparticles for reducing their harmful effects on the human body and the environment.

Important antibacterial activity against *Escherichia coli* (*E. coli* ) colonies was seen when the nanofiber soy protein was coated with silver nanoparticles [139]. In another study, silver nanoparticles were synthesized into SPI/HAP/PVA composites by a green method using UV radiation. The synthesized nanocomposites exhibited enhanced antibacterial activity against five different bacterial strains; the authors claimed that it could be a potential candidate for orthodontic applications [140]. Qu et al. demonstrated that incorporating highly dispersible TiO_2_ nanoparticles is an effective approach for enhancing the mechanical, thermal, and antimicrobial properties of zein/chitosan films. The antibacterial properties of zein/CS/TiO_2_ (0.15%) films against *E. coli*, *Salmonella Enteritidis* (*S. enteritidis)* and *Staphylococcus aureus* (*S. aureus)* under UV light conditions improved by 21.78%, 21.45%, and 26.44%, respectively [141]. 0.2% ZnONPs in SPI/ZnONPs films improved tensile strength and microbial inhibition by 231% and 16%, respectively [142].

These findings and others support the utilization of nano-sized fillers into plant proteins as part of the next generation of more complex, combinatorial medical products with improved physiochemical properties and biological activities.

### 3.3. Plant Protein-Based Electrospun Nanofibers and Films/Natural Extracts

Various plant extracts (or their refined major portions) have known drug effects and are relatively safe. Plant extracts contain bioactive compounds such as flavonoids, polyphenols, and many other biomolecules which play significant roles to treat infectious and non- infectious illnesses.

For example, *Citrullus colocynthis* extracts comprising flavonoids, alkaloids and fatty acids possess antimicrobial and antioxidant activities [143]. White tea (*Camellia sinensis*) extracts are antioxidant in nature and are of high medicinal importance for cancer therapy [144]. *Zingiber zerumbet (L.) Smith* is a wild ginger belonging to the *Zingib-eraceae* family, and is broadly used as in folk medicine, and exhibit anticancer and antibacterial activity [145]. Natural compounds isolated from green tea, catechin, blueberry, carnosine, and vitamin D3 have improved the proliferation of stem cells from bone marrow. Recently, numerous plant extracts and active constituents have been formulated as nanofibers or films for various therapeutic purposes.

For example Mariana et al. described the preparation of electrospun zein fibers, containing an IC (inclusion complex)_β-cyclodextrin (β-CD)/*Eucalyptus* essential oil (EEO) complex. The zein fibers were prepared with various concentrations of zein (20, 30 and 40%) and IC _ β-CD/EEO loading. Electrospun fibers fabricated with 30% zein displayed good uniformity in morphology. The electrospun fibers, at 24% IC _ β-CD/EEO loading, showed a 24.3% and 28.5% reduction against *S. aureus* and *Listeria monocytogenes* (*L. monocytogenes),* respectively. By increasing the concentration of IC _ β-CD/EEO, the inhibition rate was greater than that of the zeinfiber without antimicrobial function. However, the nanofibers did not display any substantial effect when tested with gram negative bacteria such as *E. coli* and *Salmonella typhimurium* (*S. typhimurium).* The electrospun IC_β-CD/EEO composite membranes are suitable for use in antimicrobial applications [146]. In another study, the *Quercetin*/gamma-cyclodextrin inclusion complex (*Quercetin*/γ-CD-IC)-encapsulated electrospun zein nanofibers were prepared by an electrospinning technique. The molar ratio of *Quercetin* and γ-CD was 1:1 in *Quercetin*/γ-CD-ICA. Nanofibers with homogenous morphology had a diameter of 750 ± 255 nm. The findings from an antioxidant activity test showed that *Quercetin*/γ-CD-IC incorporated zein nanofibers presented quite high, efficient, and quick antioxidant activity [147]. Yeum et al. reported important anti-bacterial activity of zein/*Sorghum* extract nanofibers and enhanced antioxidant capacities of zein/*Poria cocos* extract nanofibers compared to zein nanofibers only [148].

In a more recent study, *Curcumin*-loaded sodium caseinate (NaCas)-zein nanocomposite films were prepared by a pH-driven self-assembly method, which could be an alternative eco-friendly method without using alcohol or other organic solvents. The loaded *Curcumin* allowed NaCas/zein films to display a red-yellow color and antioxidant properties but showed no significant effect on their physical and mechanical properties [149]. In a similar study, *Curcumin* was introduced into konjac glucomannan (KGM)/zein nanofibers to prepare a bioactive film. The KGM/zein/*Curcumin* nanofibril films exhibited excellent antibacterial (a large inhibitory zone of 12–20 mm) and antioxidant (scavenging activity increased about 15%) activities [150].

Vahedikia et al. showed that the growth of *E. coli* and *S. aureus* was significantly inhibited by the addition of cinnamon essential oil (CEO) alone and in combination with chitosan nanoparticles (CNPs) in zein films, while CNPs-loaded zein films had no significant effect on the growth of microorganisms. Furthermore, the combination of CEO-CNPs considerably increased the tensile strength and water vapor permeability and reduced the elongation of the zein film composites [151].

In another study, *Zataria multiflora Boiss* essential oil was added to zein/sodium bentonite clay composites. Zein films containing 10% essential oil showed good antibacterial properties against *L. monocytogenes* (3.23 log) and *E. coli* (3.17 log). However, the *Zataria multiflora Boiss* essential oil caused a decrease in tensile strength and Young’s modulus of zein films [152]. Crosslinked electospun zein fibers loaded with phenolic-rich orange Chilto extracts showed antioxidant properties [97]. Olive leaf extract (OLE)-loaded zein fibers showed better proliferation and spreading of fibroblast cells on the fiber surface than pure zein [153].

In a more recent study, Xue and co-researchers prepared films from soy protein isolate gum acacia conjugates loaded with essential oils (EOs), namely oregano (OG)-EO; lemon (LM)-EO; fruit of *Amomum tsaoko Crevost et Lemaire* (ACL)-EO; and grapefruit (GF)-EOg. They showed that the physiochemical properties and biological activities of films were affected by the nature of the EO used. Amongst these films, the film comprising GF-EO exhibited the highest tensile strength, the best water vapor barrier properties and Tg compared to the films fabricated from other EOs. The films containing OG-EO and LM-EO showed the highest antibacterial activity and radical scavenging activity, respectively [154].

The addition of thyme oil to glycerol-plasticized wheat gluten protein increased in vitro antioxidant and antimicrobial properties of films. Meanwhile, it reduced tensile strength and modulus, but improved flexibility [155].

These studies and others support the utilization of natural extracts as safe materials into plant proteins for fabricating a new generation of tissue engineering scaffolds due to the their medicinal effects to control infection during and after implantation in the human body.

## 4. Plant Protein-Based Green Scaffolds for Tissue Engineering

Green chemistry is the utilization of eco-friendly and safe materials and processes to decrease the toxic and hazardous consequences of scientific research. Hence, it applies less energy, benign components, and reduces waste products generated in different synthetic procedures. The essential intention of more green methodologies and technologies is to lessen the harmful results of contaminants on living things or the environment. Besides the positive impacts of environmentally-friendly aspects, there are several deficiencies, for example more expensive costs, lack of sufficient information about used crude materials, chemicals, and methods.

Several studies have been carried out on the development of green chemistry methods concentrating on biomaterials and biocatalysis. Specifically, some scholars have studied renewable and maintainable materials for fabricating tissue engineering scaffolds. A number of natural/synthetic, biodegradable/non-biodegradable materials have been investigated to produce scaffolds without or less toxic effects in human body cells, but importantly, they found that some biodegradable scaffolds used for cardiovascular tissue regeneration have toxic decompositions and inflammatory responses and are potentially immunogenic. Therefore, researchers are attempting to find materials of lower toxicity for tissue engineering scaffolds [156].

Various methods have been applied for fabricating tissue engineering scaffolds as illustrated in Figure 5. Conventional procedures are low in cost and flexible to optimize physicochemical properties and have been applied to fabricate structures. Rapid prototyping, as a more sophisticated method, has been well applied for 3D structures and fibers, respectively, allowing the possibility of combining drugs. Significant challenges for applying these approaches are to produce benign medical and industrial products and strategically removing or reducing toxic materials and pollutants via the use of plant-based materials. Moreover, biomaterials have advantages over other materials, as they can be prepared in large quantities with similar chemical, physical, and structural characteristics for different tissue types [157].

Plant-based biomaterials can be used in various regeneration or developed scaffolds. The important edge of plant-based scaffolds is that they can be easily assembled and manipulated; they are renewable, simple to mass produce, and are low in cost and thus should be studied in animal models [157]. Moreover, such scaffolds display promising tissue compatibility, and the remaining scaffold materials during decomposition do not harm the adjacent tissues [157]. Although toxicity is an improbable problem it has potential for immune reactions if such scaffolds are inserted into a mammal. Regardless of significant advances in the production of bio-based scaffolds for tissue regeneration, nutrient transfer in complicated engineered human tissues is still a key challenge.

### 4.1. Soy Protein

Soy protein is a spherical protein isolated from soybeans and it has long-term storage stability. Soybeans comprise about 38% proteins, 30% carbohydrates, 18% oil, and 14% moisture and minerals [158]. Soy protein is prepared through the removal of oil and carbohydrates. Soy protein exists in three forms as a protein source, namely soy flour, soy protein concentrate, and soy protein isolate (SPI), and their content ranges from 50% to 90% [159]. The soy protein isolate is the most purified form of soybean proteins, containing 90% or more protein. The soy protein contains 20% acidic amino acids, which involves glutamic acid and aspartic, non-polar amino acids: valine, alanine, and leucine, and 18% basic amino acids residues, which include lysine, arginine, and the non-polar residues: glycine and cysteine [160]. Soy protein with a globular shape is more stable to hydrolysis than coiled or helical structures [161]. Therefore, when the soy protein is dipped into various pH buffer solutions, it can function as a polyanion or polycation. The isoelectric point of soy protein is around 4.8, and its solubility and swelling is low at this pH [162]. Nevertheless, the swelling ratio can be promoted by changing the pH of the buffer solution. In addition to the pH, both temperature and heating time are other effective factors which can change the dissolution and solution viscosity of SPI in water. In some recent studies, for instance, to produce soy protein fibers via electrospinning techniques, all of the mentioned conditions above have been exploited [163,164]. Moreover, when an electric field is applied, a soy protein hydrogel could be bended either toward the cathode (pH > 6) or the anode (pH < 6), which depends on the pH of the solution [119]. The SPI hydrogel generally shows desired electroactive actions under potent acidic (pH = 2m − 3) or basic (pH = 11 − 12) solutions, and such hydrogels exhibit high potential for actuator and microsensor applications, specifically for biomedical applications [165].

Soy protein, due to its biocompatibility, biochemical similarity and long-term storage stability to the natural components of the ECM has strong merit for tissue engineering applications [166]. However, the efficacy of natural polymers is limited by poor mechanical properties and rapid biodegradability. These disadvantages can be overcome by a post-spinning crosslinking procedure with proper crosslinkers, or by blending natural polymers with a biocompatible synthetic polymer, possibly resulting in an ideal scaffold for tissue engineering applications. However, as with the above approaches, the use of synthetic polymers may decrease environmental friendliness. Thus, a major disadvantage of using natural proteins in regenerative medicine is the requirement to use synthetic polymers which decrease the enthusiasm of using natural proteins in the first place. As given in Table 4, the mechanical properties of soy protein fibers are comparable with the properties of other non-crosslinked plant protein fibers.

The application of soy protein to create various types of scaffolds for tissue engineering applications is discussed in the following sections.

#### 4.1.1. Soy Protein Porous Scaffolds

Many research groups have fabricated soy protein porous scaffolds for tissue engineering applications. In one study, a cross-linked porous scaffold based on a chitosan-soy protein blend system via combining a sol-gel procedure with freeze-drying was prepared [169]. Tetraethyl orthosilicate (TEOS) was used as a cross-linker to increase the interaction at the interface between the two polymers, and thus enhanced mechanical stability and degradation rate, as well as the surface energy of the products. It was also shown that TEOS enhanced porosity, with pore sizes ranging between 50 and 350 µm and as well improved the interconnectivity among pores in the scaffold. Water uptake of the chitosan-soy-TEOS hybrids increased when compared with chitosan-soy owing to the higher material porosity. Moreover, the presence of silanol groups may compel a mineral-type apatite surface in physiological environments [170], which is related to silanol groups for bone tissue engineering applications [171]. This scaffold has been considered for cartilage tissue engineering applications.

Luo et al. [172] prepared porous membranes based on a cellulose-SPI blend system for tissue engineering applications. The membranes presented higher mechanical stability, and better cell development both in vitro and in vivo than the original cellulose membranes. The pore size on the surfaces and in the cross-sections of the membranes was enlarged with an increase of SPI concentration. In fact, the incorporation of SPI changed the microstructure of the pure cellulose scaffold leading to enhanced mechanical properties, in vivo biocompatibility as well as biodegradability. However, due to the small size of the pores, human umbilical vein endothelial cells (ECV304) were seen to adhere merely to the surface and barely entered into the scaffolds. In a similar study, SPI-cellulose sponges were fabricated through a freeze drying method [173]. The results showed that the porous scaffolds were physically robust and physiologically biocompatible to grow cells in both in vitro and in vivo experiments as compared to an original cellulose scaffold. It was seen that L929 fibroblast cells adhered, distributed, and proliferated greater in the cellulose-SPI membranes and showed higher cell viability as compared to the original cellulose sponges [173]. Furthermore, SPI-containing sponges implanted in rats showed better in vivo biocompatibility and biodegradability than the original cellulose sponge. This was due to the combination of SPI into cellulose and to the freeze-drying procedure which formed large pores and thin pore walls in the composite sponges, promoting the migration of cells and tissue into the sponges, leading to gradual fusing with the implant. The authors claimed that such scaffolds are suitable for bone and cartilage tissue regeneration.

In another recent study, a cellulose-SPI porous membrane was used as a nerve conduit to reconstruct and mend sciatic nerve defects in rats [174]. In this research, cellulose/SPI membranes containing 30% SPI was used, because it not only displayed a porous structure, but also showed proper mechanical properties in the wet state. The researchers constructed three types of nerve guide tubes: one from SPI-modified cellulose membranes (CSC) as a hollow conduit, a second one from CSC joined with Schwann cells (SCs) as seeded cells (CSSC), and the last one from CSSC combined with pyrroloquinoline quinone (PQQ) as a nerve growth factor (CSSPC). It was reported that the CSSPC conduit could greatly restore and reconstruct the nerve structure and muscle functions in comparison with the CSSC and CSC groups because of inclusive participation from the hollow CSC conduit, SCs, and PQQ. The reconstructed nerve in the hollow CSSPC conduit was directly seen, owing to its great transparency, which was an advantage over previously reported regenerative nerve fibers [175]. The authors proposed that CSSPC may have a potential use as nerve guide tubes in nerve tissue engineering.

Recently, Zhao et al. [176] used hydroxyethyl cellulose (HEC) as a water-dissolved cellulose instead of conventional cellulose to blend with soy protein isolates for improving the in vivo biodegradability of such composite scaffolds. In this study, SPI was mixed with HEC and epichlorohydrin (ECH) was utilized to cross-link the two polymers. Results showed that the films were biodegradable in vivo and the degradation rate was controlled through changing SPI concentration. The composite films with an SPI content higher than 30% showed improved mechanical, water resistance properties and suitable biodegradability. Furthermore, HEC-SPI films were biocompatible and presented good L929 fibroblast cell adherence, spreading, and proliferation in an in vivo study, making the composite film suitable for medical applications such as tissue regeneration.

Guan et al. prepared robust soy protein scaffolds with directional freezing and freeze-drying methods. Directional freezing and freeze-drying methods are much more environmentally friendly than using some of the aforementioned solvents. Soy protein solutions were properly prepared by using guanidine hydrochloride and dithiothreitol. The scaffolds prepared by directional freezing had anisotropic morphological features, which were controlled via a combination of the solution concentration and freezing rate. It was demonstrated that by increasing both the concentration and freezing rates, the diameter of the cells to the freezing axis decreased. Concentration had the most control over scaffold morphology especially with an increasing density. The development of scaffold morphology started from its fibrillar pillars, which extended to become layers which were then developed into frequently spaced ridges normal to the layers, which finally bonded to create an extremely anisotropic foam structure [177].

In other work, Chien and Shah [178] described the fabrication of porous 3D scaffolds by using soy protein modified with a heat treatment and enzymatic crosslinking using transglutaminase via a freeze drying method. The scaffolds had uneven surfaces, irregular pores with size distributions ranging from 10 to 125 µm, <5% moisture content and compressive moduli in the range of 50 to 100 Pa. Enzyme treatment enhanced the stability and degradation period, while it did not modify the mechanical properties of the scaffolds. It was also shown that scaffolds containing 5.0 wt% soy protein degraded faster than 3.0 wt% soy protein scaffolds in a PBS solution. In fact, a higher interaction between amine (-NH_2_) and carboxyl (-COOH) groups exist in soy protein with the salts in PBS easily leading to degradation of the amino acid chains. However, all of the prepared scaffolds were strongly sufficient to support the viability of human mesenchymal stem cells (hMSCs), but alterations in scaffold degradation changed the development and morphology of the attached hMSCs. The biological experiments showed that cell proliferation did not occur in the 5.0 wt% SPI scaffolds, but the 3.0 wt% SPI group with one unit of transglutaminase crosslinked, enhanced cell spreading with cells integrating into the scaffold after just 2 weeks. These results showed that the fabricated porous scaffolds have potential for numerous tissue regeneration applications.

Chein et al. fabricated 3D porous soy protein scaffolds via 3D bioplotting, which is a quick and efficient production technique. A major benefit of this fabrication approach is the capability to control pore structure and geometries essential for tissue regeneration. In this work, to prepare the scaffolds, a slurry solution comprising of 20.0 wt%soy protein, 4.0 wt% glycerol, and 7.5 mM dithiothreitol was transferred into a 3D bioplotter whereas the flow rate was adjusted to 0.007260 ± 0002 g/s at room temperature. The scaffolds were treated with diverse curing processes and classified into four groups: (i) nontreated (NT), (ii) dehydrothermal treated (DHT), (iii) freeze-dried and dehydrothermal treated (FD-DHT), and (iv) chemically cross-linked using 1-ethyl-3-(3 dimethylaminopropyl) carbodiimide (EDC). Scaffold groups were sorted from the least to most strong in the following order: NT, FD-DHT, DHT, and EDC. The EDC crosslinked scaffolds were the strongest scaffolds with a modulus of ~4 kPa. The highest seeding efficacy of hMSCs was seen for thermally treated and non-treated scaffolds; however, all the scaffolds supported the viability and growth of hMSCs over time [179].

Chien et al. [180] prepared soy protein porous scaffolds using two different methods, namely freeze drying and 3D printing. 1.0 wt% and 3.0 wt% soy freeze-dried scaffolds showed the largest volume percentage of pore sizes, ~60 µm and 16 µm, respectively. The denser bioplotted scaffolds had a higher pore volume at a pore size of 5 µm due to the channels on the printed part surfaces, and since the printed pore structure was interconnected. The acute immune reply of these two forms of scaffolds was compared when they were inserted dermally in mice. Hematoxylin and eosin (H&E) staining of tissue revealed that the freeze dried soy protein scaffolds had better cell infiltration as compared to collagen scaffolds. In contrast, cell infiltration was prevented in the denser bioplotted soy scaffolds, which caused a slower break down. The degradation rate of the scaffolds was modified in the freeze dried scaffolds containing 3.0 wt% soy protein or bioplotted soy scaffolds (20 wt%). However, the degradation rate was not changed for the 1.0 wt%freeze dried scaffolds. After 56 days, neutrophils still remained at the ulcer site, and freeze-dried scaffolds (3.0 wt%) and bioplotted scaffolds showed more inflammation. It was found that density, porosity, and degradation rate of the scaffolds did significantly affect the in vivo response. 

In another study, soy protein blends with gelatin, alginate, and pectin were assembled into porous 3D structures by involving chemical crosslinking and freeze-drying. The obtained blended structures combined adequate porosity with a great pore size and suitable interconnectivity. The best weight ratio of soy protein: natural polymer was 4:1, which offered a uniform solution and porous structure. These 3D porous scaffolds displayed promising physicochemical and biocompatibility properties and have potential for use as improved skin regeneration scaffolds [181].

In a recent study, two types of porous soybean scaffolds were fabricated, the first based on the traditional tofu manufacturing procedures, the second treated through covalent crosslinking. Both scaffolds displayed similar porous micromorphology, good cell proliferation, and cellular attachment. No noticeable inflammatory response was seen after dermal implantation tests for either material. These results confirmed that the tofu scaffolds or soybean protein scaffolds produced by tofu processing have potential in tissue engineering applications, and are much more environmentally friendly than traditional covalent crosslinking methods [182].

#### 4.1.2. Soy Protein Fibrous Scaffolds

Fibers are preferred as compared to films and other forms of scaffolds for tissue engineering because fibers more exactly mimic the natural structure of the extracellular matrix (ECM) [183], can control the development of cells, and are more robust than films.

Two techniques can be utilized for making protein fibers: the first is electrospinning, for producing nanofibers, and the second is solution and melt-spinning for making conventional fibers. The former is better for biomedical applications because of its nanofibrous structure that mimics the natural ECM environment and provides high surface area for cell adhesion and drug loading. The former developed with soy protein fibers had poor mechanical properties and were thus crosslinked and/or blended with synthetic polymers to produce fibers with enhanced properties for applications in tissue engineering; however, this reduced environmental friendliness.

Reddy and Yang [167] made 100% soy protein fibers 75 µm in diameter. These fibers showed high mechanical properties and water stability without the use of any external crosslinking agent, and in cell culture tests, fibroblasts effectively attached, grew and proliferated.

Lin et al. [161] described the preparation of soy and zein protein fibrous scaffolds by an electrospinning method for applications in skin tissue engineering. Soy protein was dissolved in 1,1,1,3,3,3 hexafluoro-2-propanol (HFP) as a solvent to prepare uniform solutions for electrospinning. Adding trace amounts of poly(ethylene oxide) (PEO) as a cross linker could help to form soy protein fibers, because soy protein cannot be electrospun into fibers by itself. Furthermore, soy/PEO fibrous scaffolds are stable in an aqueous medium, without needing to add additional cross-linkers. Such scaffolds supported the attachment and proliferation of cultured human dermal fibroblasts (HDF). The authors claimed that soy is suitable as a scaffold for organotypic skin equivalent culture, also implantable platforms for skin regeneration [161].

Ramji and Shah [164] fabricated SPI/PEO electrospun nanofibers for tissue engineering applications. It was found that the concentration of both SPI and PEO as well as the PEO molecular weight were effective for controlling fiber morphology. Electrospun fiber diameter increased after carbodiimide crosslinking but had no significant effect on porosity. PEO increased hydrophilicity of SPI. Mechanical results showed that the Young’s modulus of the crosslinked SPI/PEO fibers increased from 75 to 252 kPa when concentrations of SPI were augmented from 7.0 wt% to 12.0 wt%. Biological studies showed that the number of human mesenchymal stem cells (hMSCs) attached to 12.0 wt% SPI scaffold was significantly higher than the 7.0 wt% SPI scaffold. This is most probably due to higher SPI content providing for more protein-cell binding sites along with the greater thickness of the 12% SPI scaffolds, while little cell proliferation occurred on the 12% SPI scaffolds. This study showed that hMSCs effectively attached and proliferated on the SPI/PEO scaffolds, making such scaffolds efficient for tissue engineering applications.

In a study, soy protein was electrospun into 3D and 2D ultrafine fibrous scaffolds via dissolving the soy protein into an aqueous solvent system containing cysteine (10 wt% of soy protein) as a reducing agent [184]. Cysteine is clearly part of the family of environmentally-friendly biomolecules. A reductant was used to cleave disulfide crosslinks in soy protein and to facilitate soy protein dissolution in an aqueous solvent system. Without any external crosslinking, the soy protein scaffolds exhibited significant water stability via maintaining their fibrous morphologies for up to 28 days after incubation in PBS. In vitro studies showed that the 3D scaffolds better supported adipose derived mesenchymal stem cells spreading and adipogenic differentiation compared to the 2D scaffolds. This is most probably due to the close-packing fiber arrangement in 2D structures; moreover, cell penetration was difficult [184]. The authors proposed that 3D soy protein scaffolds could be attractive alternative for soft tissue engineering applications.

Wongkanya et al. [185] described the preparation of sodium alginate (SA), soy protein isolate (SPI), and poly(ethylene oxide) (PEO) blended nano-sized fibers encapsulated with vancomycin by an electrospinning method. The best polymer composition of the electrospinning solution was specified as 5.6/2.4/2 SA/PEO/SPI to fabricate smooth and uniform fibers with diameters ranging from 60 to 600 nm. The polymer blend composition intensely affected the fiber morphology, and subsequently, the drug release behavior. The vancomycin-loaded SA/PEO/SPI provided a slower release of drug in the initial step followed by a steady release over a longer period than the SA/PEO fibers. The nanofibers were nontoxic and biocompatible with antibacterial activity. The authors suggested that the nano-sized fibers are an attractive biomaterial for use in the biomedical area, for example, for scaffolds for tissue engineering and drug delivery systems.

In a study, SPI-based electrospun fibrous scaffolds containing nano and micron-sized 45S5 bioactive glass (BG) were prepared for tissue engineering (TE) applications. The results showed that a relative humidity over 50% can change the fiber morphology and also the uniform distribution of fibers in the mat. Manufacturing methods such as these, which involve humidity, should be extensively researched since they are environmentally safe processes. Fiber diameter decreased with incorporation of both micro and nano-sized BG. The SPI-based fibrous scaffolds have no toxic effect with similar cell viability and support MEF cell proliferation after 24 h of cultivation [186].

In a recent study, Chuysinuan et al. [187] described the preparation of nanofibrous core-sheath structured scaffolds containing tetracycline-loaded alginate/soy protein isolates (TCH-Alg/SPI) as a core and polycaprolactone (PCL) as a sheath by co-axial electrospinning. The structural stability of fibers in aqueous solutions was improved by coating hydrophobic PCL on TCH-Alg/SPI fibers as compared to bare fibrous scaffolds which quickly degraded and provided rapid drug release. In vitro studies showed that the core-sheath scaffolds were compatible and achieved great cell viability of up to 100 % in treated human dermal fibroblasts. Furthermore, such scaffolds showed antibiotic activity against pathogenic microorganisms. The authors suggested that TCH-Alg/SPI fibrous scaffolds have potential for use as temporary templates for tissue regeneration.

The mechanical properties of fibrous SPI scaffolds for skin tissue regeneration were controlled by two diverse techniques (electrospinning and wet spinning), as shown in Table 5. Electrospun fibers displayed lower mechanical properties than that of the fibers prepared by a wet spinning method.

#### 4.1.3. Soy Protein Hydrogel Scaffolds

Silva et al. [189] fabricated composite hydrogels of natural polymers, based on the blend of alginate and SPI, and the incorporation of micron sized bioactive glass (BG) particles by a sonochemical method. The BG particles were used for their mineralization potential, osteoconductive, and angiogenic properties. The addition of soy protein led to a decrease in porosity of pure alginate and improved its mechanical properties. On the other hand, the incorporation of BG particles increased the mechanical properties and also the bioactivity of the hydrogels. The composite hydrogel containing 1% (*w*/*v*) of BG particles showed the greatest in vitro bioactivity. The authors claimed that the developed composite hydrogel scaffolds are potential biomaterials for bone tissue engineering applications.

The most recent studies using soy protein-based matrices/scaffolds for tissue engineering applications are summarized in Table 6.

### 4.2. Zein Protein

Zein protein is a storage protein which is found in corn. Zein solubility is attributed to its high amount of non-polar amino acid residue content. It has an isoelectric point of 6.2. Thus, it is unsolvable in water or in phosphate buffer saline (PBS) at pH 7.4; nevertheless, it becomes solvable in the presence of a high concentration of urea, alcohol, alkaline pH (≥11), or anionic surfactants. Its molecular structure is a spiral wheel confirmation with nine homologous repeating units arranged in an anti-parallel form stabilized via hydrogen bonds [194]. Zein can be transformed into microspheres, nanoparticles, fibers, films, and composites in combination with biopolymers [195]. Because of its toughness, flexibility, water swelling, non-toxicity, and good biocompatibility and biodegradability, zein has high potential for applications in several biomedical fields. Both zein and its by-products have shown desirable cell compatibility [196,197]. In recent years, 3D zein scaffolds have been developed for tissue engineering applications. An in vitro study presented that zein scaffolds could promote the attachment, proliferation, and osteoblastic specialization from human mesenchymal stem cells (hMSCs) [198]. Moreover, porous zein scaffolds degrade entirely within 8 months [199]. The application of zein to produce various types of scaffolds for tissue engineering is discussed in the following section.

#### 4.2.1. Zein Porous Scaffolds

Particle leaching is one of the most extensively used approaches to achieve a controlled porosity size in zein scaffolds. Several research groups have fabricated zein porous scaffolds by means of a particle leaching process alone or in combination with other methods. For example, in one study, the protein zein was made into a porous scaffold using a salt-leaching process suitable for bone applications. The zein scaffolds had a high porosity of 75.3%–79.0%, good pore interconnectivity, high mechanical properties and a degradation rate of 89% using pepsin and 36% using collagenase after incubation for 14 days in vitro. Such scaffolds supported the adhesion, growth, proliferation, and differentiation of hMSCs [200].

In a similar study, a zein porous scaffold was produced by means of a solvent casting/particulate leaching process. The scaffold with good biocompatibility, proper porosity (64.1%–78.0%), and well interconnectivity was suitable for the development of periodontal ligament cells (PDLCs) [201]. In another study, Wu et al. [202] created porous scaffolds of zein/PCL composites through a solvent casting/particulate leaching technique to regenerate bone. The obtained porous biocomposite scaffolds had high porosity (~70%) and well-interconnected network. Zein/PCL scaffolds showed better hydrophobicity than PCL scaffolds. Moreover, zein/PCL scaffolds decomposed faster after incubation in PBS for 28 days as compared to the PCL scaffolds. This study showed that the degradation rate could be tailored by altering the zein concentration in the composite to match the rate of tissue regeneration [202].

In another study, zein tubes were fabricated via a dipping-leaching method and were applied to restore damaged rat sciatic nerves with a 10 mm imperfection. 3D zein conduits were fabricated with/without pores, and with/without microtubes in the lumen of conduits. Both the porous zein conduits and the microtubes had a porosity of about 80%. The pore sizes on both the external surface and cross-section of the conduit chiefly ranged from 1 to 30 mm, hence fulfilling the requirements of the ideal nerve conduit. The 3D porous zein conduits containing microtubes showed a good balance between mechanical properties and decomposition without breakdown and compression. These conduits were completely degraded within two months without causing a physical blockage for nerve regeneration [203].

#### 4.2.2. Zein Fibrous Scaffolds

Zein is the most extensively used plant protein and can be easily assembled into nanofibers using an electrospinning process because of its solubility in ethanol, which is otherwise not expected with insoluble proteins [204,205].

However, zein has problems commonly associated with other protein-based biomaterials, for example, low mechanical properties and poor structural stability in an aqueous environment, particularly when fabricated into electrospun fibers. Hence, attempts have been carried out to enhance the mechanical and water stability of zein fibers by blending them with synthetic polymers or using toxic chemical crosslinking agents (for instance, glutaraldehyde, formaldehyde, etc.). Glutaraldehyde is toxic and other crosslinking agents possess insufficient efficiency. Thus, again, we see the need for the improved development of environmentally-friendly methods to enhance mechanical stability and water stability of natural proteins. Jiang and Yang utilized citric acid as a cross linker for electrospun fibers of zein [32]. The authors found that the cross-linked zein fibers preserved their morphological structure even after incubation at 37 °C in PBS for up to 15 days. However, while the crosslinking effects of citric acid are higher than most existing non-toxic crosslinkers, they are still too low for mini molecule aldehydes. In another study, 3D ultrafine fibrous zein scaffolds were crosslinked with strong non-cytotoxic poly aldehydes obtained from sucrose. It was found that oxidized sucrose cross-linked 3D zein scaffolds presented notable water stability as compared to those cross-linked with citric acid. However, the mechanical properties and protein sorption of both cross-linked samples were not significantly different and their biological activities were similar [206].

Zhang et al. [207] fabricated zein fibrous membranes reinforced with bone matrix-mimic HA nanoparticles via electrospinning. The zein and HA solution was mixed through a magnetic stirrer (Method I) or ultrasonic power (Method II). The HA was homogenously distributed in the membranes electrospun using Method II. The average fiber diameter increased gradually with the increase in concentration of HA particles, but the increment for the nanocomposite fibers electrospun by Method II was smaller than that of the nanocomposite fibers electrospun by Method I. The wettability of zein/HA fibrous membranes was enhanced by a magnetic stirrer, while it exhibited no adverse influence on tensile strength, and both membranes showed desirable mechanical properties. The cells seeded on the zein/HA scaffold were electrospun using a magnetic stirrer containing 5.0 wt% HA and displayed considerably higher proliferation compared to those seeded on the control zein scaffolds on the seventh day. This study showed that the zein/HA nanofibrous membranes fabricated with high biological efficiency are more interesting for bone tissue engineering applications than respective monolithic materials [207].

In another study, Figueira et al. produced a bilayered electrospun membrane using the electrospinning method for skin tissue regeneration. Hyaluronic acid and polycaprolactone (HA PCL) were used to fabricate the upper layer in order to provide mechanical support and similarly to function as a physical obstruction against outside threats. The bottom layer of the membranes contained chitosan and zein which was loaded with salicylic acid (CS ZN SA) in order to offer antimicrobial and anti-inflammatory activity to this layer. The prepared electrospun membranes showed good mechanical properties, controlled water loss and a proper drug release profile. HA PCL and CS ZN SA layers presented a highly porous 3D nanofiber network composed of randomly oriented fibers with diameters of 472 ± 192 nm and 530 ± 180 nm, respectively, which provided an ideal microenvironment for cell recruiting/seeding, attachment, proliferation, differentiation, and eventually improved skin tissue regeneration. The in vitro studies also showed that the membranes were non-cytotoxic for human fibroblast cells and provided a 3D polymeric support to allow for cell adhesion, distribution, and proliferation [208].

In a recent study, zein/poly (3-hydroxybutyrate-co-4-hydroxybutyrate) [P(3HBco-4HB)] blended fiber scaffolds were fabricated via an electrospinning technique using N,N-Dimethylformamide (DMF) as a co-solvent. The fibrous scaffolds showed high porosity (85 ± 2.1%–92 ± 2.6%), good interconnected pores and a large specific surface area which were assembled from ultrafine fibers with diameters in the range of 60–650 nm. The tensile strength and elongation at break of the blended fibrous scaffold was enhanced about 50% and 400%, respectively, as the content of P(3HBco-4HB) increased from 20% to 80%. Based on vitro assays, it was found that the blended scaffolds were noncytotoxic for NIH3T3 fibroblast cells and MG-63 osteoblast cells and supported cell attachment, distribution, and proliferation. The good physiochemical and biological properties of electrospun fibrous scaffolds make them attractive for use in tissue engineering applications [209].

Furthermore, Liao et al. [210] prepared micro/nano fibrous PCL/zein- calcium lactate (CL) scaffolds by means of an advanced two-nozzle electrospinning technique. Results showed that the addition of the CL (5.0%) to the PCL/zein fibers could enhance the tensile strength, wettability, and biological activity of the composite films with a strong potential for bone tissue engineering applications.

In another study, homogenous electrospun fibers were made by blending poly(glycerol sebacate) (PGS) and zein in acetic acid, as a nontoxic solvent. These PGS–zein fibrous structures showed high mechanical properties, good water uptake characteristics, and high porosity. The pore size of the PGS–zein fiber mats ranged between 750 and 850 nm. Mechanical properties and water uptake characteristics of PGS–zein fibers increased as did the amount of zein augmented in the fibrous blends. The obtained porosity and water uptake characteristics indicated that the fibrous PGS–zein scaffolds are in principle suitable for enabling cell adhesion. The researchers suggested that the PGS–zein fibrous structures are attractive biomaterials suitable for cardiac patch applications [211]. In addition to the above work, where 6:1, 5:1, and 4:1 ratios of zein-PGS were studied, a similar study focused on the production of 6:1, 3:1, and 1:1 ratios of zein-PGS, as well as zein-mildly cross-linked PGS blends in less toxic solvents such as ethanol and acetic acid [212]. The authors presented that adding PGS to zein and its increase in content decreased fiber diameters from around 0.3 µm to 90 nm. Mechanical tests displayed that the incorporation of PGS into zein did not considerably affect the Young’s modulus, but the final failure strain and tensile stress was enhanced four- and seven-fold, respectively, as compared to the control zein fiber mats. The fibers were degraded in PBS after one day of immersion. A better aqueous stability was observed when zein was cross-linked using the zero-length cross-linking reagents *N*-(3-Dimethylaminopropyl)-*Ń*-ethylcarbodiimide (EDC)/N-Hydroxysuccinimide (NHS). The authors claimed that the fibers have potential for use in soft tissue engineering applications.

In another study, a calcium phosphate cement (CPC) surface was coated by porous zein/PLLA coaxial nanofibers by using an auxiliary receiver and based on electrostatic spinning technology and a salting out process. The composite not only preserved the superlative chemical and physical performance of the original CPC, but it also led to the creation of a hydrophilic surface with proper mechanical properties, superior biocompatibility, and a notable density of viable cells. It also provided a suitable environment for cell attachment and proliferation. The researchers proposed that the 3D coating of porous zein/PLLA coaxial nanofiber membranes on CPC surface composites represents a potential scaffold for use in bone tissue engineering applications [213].

In recent studies, HAP nanoparticles in combination with zein or the blending of zein with natural/synthetic polymers have been used to fabricate composite scaffolds for tissue engineering applications. For example, Pedram Rad et al. produced PCL/zein/gum arabic (GA) porous nanofiber scaffolds in different concentrations and ratios by an electrospinning method. A PCL polymer was used for elasticity, strength, and time of scaffold degradability. PCL/zein/GA scaffolds displayed high hydrophilic properties, high porosity (~80%), tensile strength of 1.36–3.0 MPa and an elongation at break of 19.13%–44.06% suitable for skin tissue engineering. Moreover, in vitro assays revealed favorable L929 cell adhesion and proliferation [214]. This research group in other work showed that loading *Calendula officinalis* extract on PCL/zein/GA composite scaffolds can improve the antibacterial activity and biocompatibility of PCL/zein/GA scaffolds [215]. In another study, a zein/poly (sodium 4-styrene sulfonate)-modified hydroxyapatite nanoparticle (PSS-modified HAP) composite scaffold was prepared for bone tissue engineering applications. The surface of PSS-modified HAP nanoparticles was loaded with a vancomycin drug to control infection during implantation. The results showed that with increasing HAP concentration in the scaffold from 0 to 10 wt%, a reduction in porosity (from 297 ± 1.314 to 209 ± 1.016), the size of pores from 86.19 ± 1.64 to 75.01 ± 0.96, microns), hydrophilicity (from 85.45 ± 3.67 to 61.46 ± 1.37 contact angles, degrees), and scaffold degradation rate was observed. Furthermore, a significant decrease in strength to 30.8% and compression modulus to 76.7% was seen. MG-63 cell viability tests exhibited >90% viability of cells in scaffolds containing HAP nanoparticles [216]. Hydroxyapatite can cause an enhancement in the bioactivity of the scaffold, which is a factor in increasing the biological properties and viability of the cells in the scaffold. Lian et al. (2019) prepared a HAP/zein composite membrane for bone regeneration. They showed that for the higher content of HAP (90 wt%), nanowires contributed to an improved fibrous microstructure and enhanced mechanical properties (about 40%) and water absorption (about 80%) and improved the adhesion, proliferation, and osteogenic differentiation of MSCs [217]. In a similar study, a porous scaffold of zein/chitosan/nanohydroxyapatite (nHAP) was prepared using a freeze-drying procedure. The results showed that the incorporation of nHAP enhanced the mechanical properties, thermal stability and protein adsorption of/on the composite scaffolds. The scaffold showed good MG-63 cell adhesion, growth, and proliferation, making them capable for tissue regeneration such as bone repair [218].

Studies using zein protein-based matrices/scaffolds for tissue engineering applications are summarized in Table 7.

### 4.3. Wheat Gluten (Gliadin, Glutenin) Protein

Gluten protein represents ~85% of wheat proteins and is made up of gliadin and glutenin proteins. About 50–55% of the proteins are monomeric gliadins, with the remainder disulfide crosslinked polypeptides that make the polymeric glutenin portion. The gluten proteins are mainly hydrophobic in nature, and thus, are insoluble in water. Gluten’s solubility can be enhanced by different approaches, such as acid or alkali modification, enzymic hydrolysis of the peptide bonds and physical treatment. Wheat gluten, due to its good water and heat stability, remarkable elasticity, and good degradability properties, is promising as a fibrous biomaterial for biomedical applications (such as implants and tissue engineering scaffolds).

#### Wheat Gluten Scaffolds

In a study, wheat gluten proteins (gliadin and glutenin) were fabricated into membranes for potential applications such as scaffolds for tissue engineering. The films presented significantly varied behavior in terms of decomposition in water and the capability to support the adhesion and proliferation of fibroblasts. A weight loss of 90% and 50% were seen for wheat glutenin and gliadin, respectively, when immersed in water at a pH 7.4 at 37 °C for 15 days. Gliadin was cytotoxic to fibroblast cells, but gliadin-and-starch-free glutenin was cytocompatible and showed much greater adhesion and proliferation of fibroblast cells in comparison with poly(lactic acid) (PLA) films [12].

A high crosslinking degree in wheat gluten makes it insoluble in typical solvents, and thus hardly has to be electrospun or freeze dried by itself. To date, fibrous structures containing wheat gluten have been successfully fabricated by involving either the mixing with synthetic polymers or being of cytotoxic proteins after the hydrolysis of wheat gluten, which again transitions such materials from environmentally-friendly to environmentally unfriendly.

Woerdeman et al. fabricated electrospun wheat gluten by the blending of wheat gluten and gliadin, which was, most probably, the first plant protein to have been electrospun [28]. Soluble wheat gluten has also been blended with PVA and then electrospun [234]. Nevertheless, gliadin showed cytotoxicity [235], while PVA could notably decrease the propensity of scaffold-cell interactions. The disulfide crosslinks that exist in wheat gluten should be broken while the backbone should be retained before dissolving wheat gluten and being electrospun into 3D structures [236]. Xu et al. dissolved wheat glutenin in an aqueous solvent containing sodium dodecyl sulfate (SDS) and then electrospun them into 3D fibrous scaffolds to mimic the natural extracellular matrices of soft tissues [237]. The 3D fibrous wheat gluten scaffolds presented superior support for the proliferation and adipogenic differentiation of adipose derived mesenchymal stem cells in comparison with 2D ultrafine fibrous wheat gluten and commercial 3D non-fibrous scaffolds.

### 4.4. Camelina Scaffolds

Camelina protein (CP) is biocompatible and biodegradable which has the potential to develop into water stable scaffolds for biomedical applications [238]. Hitherto in a few studies, camelina protein has been utilized to produce scaffolds for tissue regeneration applications.

For example, in a study, Zhao et al. [238] fabricated CP films for tissue engineering applications. CP was dissolved in a solvent system comprised of cysteine/urea and sodium dodecyl sulfate (SDS). CP films showed a weight loss of ~12% after incubation in PBS for 7 days. CP films could support seeded cells for up to 18 days, without cleaving. The properties of CP films were dependent on production conditions (like aging time and SDS concentration). Aging time and SDS affected film stiffness while clearly the former is an environmentally-friendly process. Mouse fibroblasts seeded on the CP films presented a greater degree of metabolic activity in comparison with collagen films under similar culture conditions. The author proposed that these films could be suitable for tissue engineering.

### 4.5. Aloe Vera (AV) Scaffolds

Aloe vera as a leaf protein concentrate is a rich source of amino acids (the building blocks of proteins) naturally with the eight essentials amino acids needed in the body [239]. Recently, aloe vera has been used to improve the structure, composition, biodegradability and cell proliferation on scaffolds.

In a recent study, adding different concentrations (0.1–0.5%) of aloe vera to collagen–chitosan (COL-CS) scaffolds enhanced thermal stability as well as hydrophilicity and reduced tensile properties of the scaffolds. The scaffold exhibited enhanced growth and proliferation of fibroblasts (3T3L1) without showing any toxicity [240].

Kim et al. prepared transparent ultrathin film scaffolds with an aloe vera (AV) gel and silk fibroin (SF) for corneal endothelial cells (CECs) [241]. Field emission scanning electron microscopy (FESEM) observations showed that a critical morphology of CECs was formed on the AV/SF blend rather than in the scaffold with pure SF. Incorporation of a small quantity of aloe vera gel improved cell viability and sustained its functions well. The scaffolds were applied for transplantation into rabbit eyes. The scaffolds attached to the surface of the corneal stroma and integrated with surrounding corneal tissue without a major inflammatory reaction. The authors claimed that AV blended SF film scaffolds might be an appropriate substitute for alternative corneal grafts for transplantation.

Selvakumar et al. fabricated guided bone regeneration (GBR) with an anti-infective electrospun scaffold by ornamenting segmented polyurethane (SPU) with two-dimensional aloe vera wrapped mesoporous hydroxyapatite (Al-mHA) nanorods [242]. The Al-mHA frame was introduced into an unprecedented SPU matrix based on combinatorial soft segments of poly(ε-caprolactone) (PCL), poly(ethylene carbonate) (PEC), and poly(dimethylsiloxane) (PDMS), by an in situ method followed by electrospinning to fabricate scaffolds. The scaffolds showed a remarkable improvement in mechanical properties (175%), biodegradation, and biocompatibility against osteoblast-like MG63 cells (in vitro), with favorable antimicrobial activity against various human pathogens. These scaffolds were implanted in rabbits as an animal model. Early cartilage formation, endochondral ossification, and rapid bone healing at 4 weeks were found in the defects filled with an Al-mHA ornamented scaffold compared to pristine SPU scaffolds. This study showed the advantages of an aloe vera wrapped mHA frame in promoting the osteoblast phenotype with microbial protection for potential GBR applications.

In another study, polycaprolactone (PCL) containing 5.0 and 10.0 wt % lyophilized powder of aloe vera was electrospun into nanoscale fiber mats compared with a PCL/collagen blend for dermal substitutes [243]. The average diameters of PCL-AV 5.0 % and PCL-AV 10.0% were in the range of 264 ± 46 and 215 ± 63 nm, respectively. They found that PCL-AV 10% nanofiber scaffolds with finer fiber structures improved hydrophilicity, tensile strength (6.28 MPa), and Young’s modulus (16.11 MPa), all desirable for skin tissue engineering. It was also found that a PCL-AV 10 % nanofibrous matrix favored cell proliferation compared to other scaffolds which almost increased linearly by 17.79% and 21.28% compared to PCL on the sixth and ninth day. Secretion of collagen and F-actin expression were significantly increased in PCL-AV 10% scaffolds compared to other nanofibrous scaffolds. The results demonstrated that the PCL-AV 10% nanofibrous scaffold is a potential biomaterial for skin tissue regeneration.

López Angulo et al. prepared scaffolds based on gelatin and chitosan (G-CH) combined with a small quantity of aloe vera and a snail mucus blend for skin replacement. Homogeneous networks and interconnected porous structures in the composite scaffold were observed by scanning electron microscopy (SEM) after crosslinking with glutaraldehyde and freeze-drying. The addition of aloe vera and snail mucus increased the average pore size from 119 ± 38 µm to 207 ± 61 µm and enlarged porosity from 80.2% to 94.0% causing modifications in the pore architecture. Scaffold porosity, pore size, and the entire pore structure all have significant results upon tissue development and penetration into biomaterial structures. Interconnecting pores enable cell loading into scaffold, while the enlarged interior surface area provides opportunities for cell attachment and spreading. The results showed that the incorporation of a small quantity of aloe vera and snail mucus blend could improve flexibility, water absorption capacity, biodegradability, and cell response of the G-CH scaffolds, making them suitable for numerous tissue regeneration applications [244].

In one study, aloe vera and silk fibroin (SF) with 4% hydroxyapatite (HA) as a bioactive agent was added to poly (lactic acid-co-caprolactone) (PLACL) to prepare PLACL-AV-SF-HA (4%) nano-fibrous scaffolds via electrospinning for bone tissue engineering applications. Human mesenchymal stem cells (hMSCs) cultured on PLACL-AV-SF-HA (4%) nano-fibrous scaffolds exhibited important increases in cell osteogenic proliferation, differentiation, expression, and mineral deposition in comparison with different controls. It was found that the synergistic influence of the osteo-inductive property of aloe vera accompanied by osteo-conductive hydroxyapatite increased the differentiation and biological performance of hMSCs to osteoblasts with the proper mechanical properties offered via skin fibroin, verifying PLACL-AV-SF-HA (4%) to be an extremely suitable scaffold for bone tissue regeneration [245].

In another study conducted by Carter et al., aloe vera was blended with polycaprolactone (PCL) to fabricate nano-fibrous guided tissue regeneration (GTR) membranes by electrospinning. PCL/AV nano-fibrous membranes with proportions from 100/00 to 70/30 displayed good evenness in fiber morphology and proper mechanical properties, and maintained the integrity of their fibrous structure in aqueous solutions. The PCL/AV membranes supported 3T3 cell viability and could be a potential candidate for guided tissue regeneration (GTR) therapy [246].

Isfandiary et al [247] fabricated collagen-chitosan-AV composite scaffolds by a freeze drying method for healing burned skin tissue. The preparation of scaffolds was carried out by dissolving collagen-chitosan (1:1) in 0.05 M acetic acid; then, consequent variants of AV (0%, 0.1%, 0.15%, 0.2%, and 0.25%) were added into solution. Results showed that a composite with 0.2% AV was a strong potential candidate to serve as improved scaffolds for burned skin tissue applications [247].

## 5. Conclusions

Plant proteins are cytocompatible and biodegradable materials that can be fabricated into micro/nanoparticles, fibers, porous structures, hydrogels, and composites with good properties suitable for tissue regeneration. However, chemical and physical treatments are needed to obtain the required physiochemical properties and biological activities for plant protein-based biomaterials for tissue engineering applications. While the physical processes outlined here are attractive and environmentally-friendly, many of the chemical processes developed to date by researchers are not, and, thus, remove the attraction for using plant proteins in the first place. As reviewed in this article, plant proteins are being increasingly considered for several tissues engineering applications including bone and cartilage engineering, skin tissue regeneration, and nerve and cardiac regeneration. They represent a new generation of green technologies where Mother Nature herself has created such outstanding tissue engineering constructs. At a time when it seems every other field has embraced green technologies, the present article highlights some (albeit much more is needed) of the intersection between regenerative medicine and green medicine. Such tissue engineering applications of plant based materials were reviewed in the present article and accessible information from the literature was discussed, mostly associated to the properties, processing, and biological behavior of plant protein-based materials. Between the plant proteins, zein has been the most widely investigated for the fabrication of porous structures, fibers, and composite biomaterial scaffolds for several tissue engineering applications. The emerging uses of plant proteins in tissue engineering and related developments made over the last decade highlight that this area will grow toward an increasing use of plant-proteins based scaffolds in tissue engineering segments, especially if environmentally-friendly approaches are discovered to increase their strength and stability.

## 6. Future Perspectives

Large opportunities still exist to fabricate novel kinds of plant protein-based scaffolds with attractive properties, which could be used for tissue regeneration. For example, the insertion of fillers (cellulose nanocrystals and nanofibrils, inorganic and metallic particles, carbon dots, carbon nanotubes, graphene oxide, and so on) into plant protein matrices has been shown to be an attractive procedure to simulate the natural characteristics of electro-active and load-tolerating tissues in the human body. The involvement of filler materials not only enhances their select properties (such as electrical properties), but also promotes mechanical properties and biological behaviors in relation to the original plant protein-based scaffolds. Thus, the design and development of plant protein–based scaffolds with improved, mechanical, electrical, and biological properties are continually expected.

More studies on the chemical treatment of plant protein structures by using different chemical and biological compounds as well as physical treatments are required to increase physicochemical properties and biological activities, such as cell adhesion, distribution, proliferation, and differentiation on plant protein based scaffolds. However, it is required to preserve the inherent attractive properties of plant proteins after modification.

3D printing is a novel production technique that has aided researchers to fabricate favorable structures that simulate the natural biological environments. Hitherto, a few works have been reported relevant to utilizing plant proteins to produce scaffolds by 3D printing. However, solvability in aqueous solvents, processability, and stability of plant proteins are likely important difficulties. Furthermore, structural restrictions will remain the main challenge that will encourage upcoming studies, especially those approaches that use green chemistry.

The design of scaffolds should be in such a manner that they are able to save their morphological structure, physiochemical, and biological properties after being implanted in the body; they must also be compatible with in vivo systems. These basic principles should be strongly considered in producing a scaffold to substitute the ECM and to repair or regenerate an injured organ or tissue. Nonetheless, the use of environmentally friendly plants in regenerative medicine has started and promises to be a bright future to simultaneously preserve the environment and improve tissue growth above that of traditional synthetic polymers.

## Figures and Tables

**Figure 1 biomolecules-09-00619-f001:**
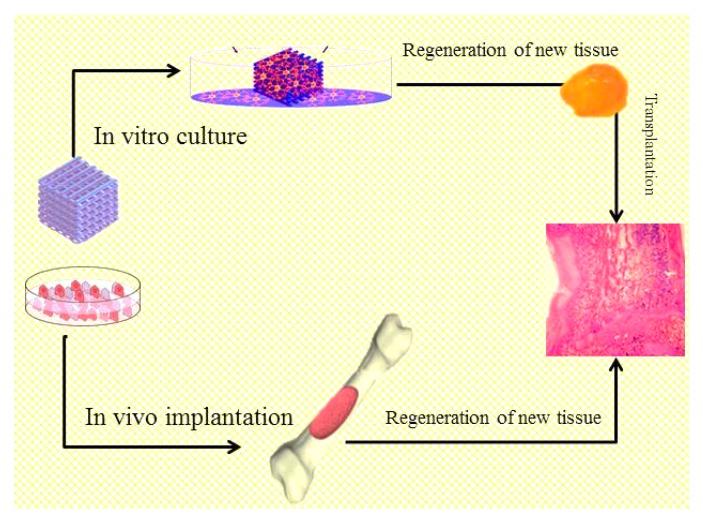
In vitro culture and in vivo implantation of cell-seeded scaffolds to generate new tissue.

**Figure 2 biomolecules-09-00619-f002:**
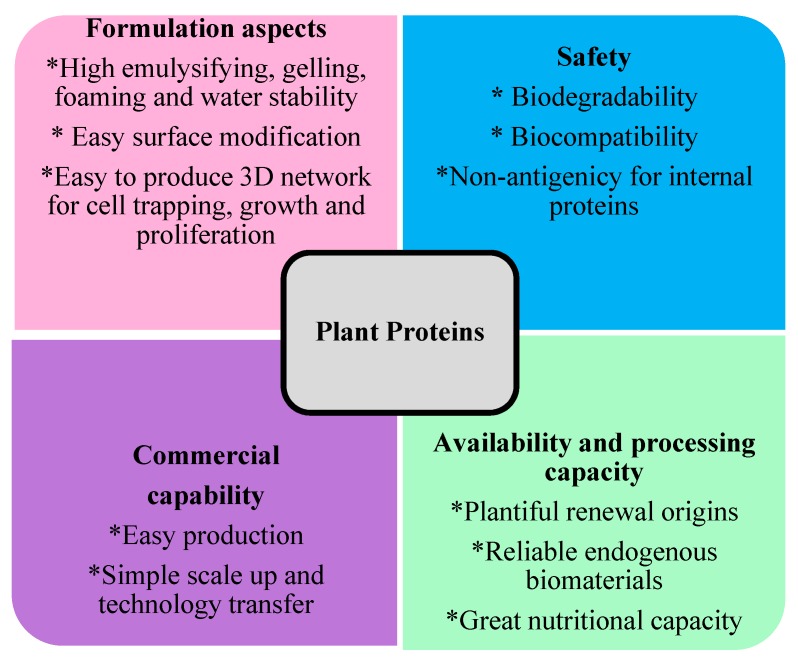
Attractive aspects of plant proteins for biomedical applications. 3D: three-dimensional.

**Figure 3 biomolecules-09-00619-f003:**
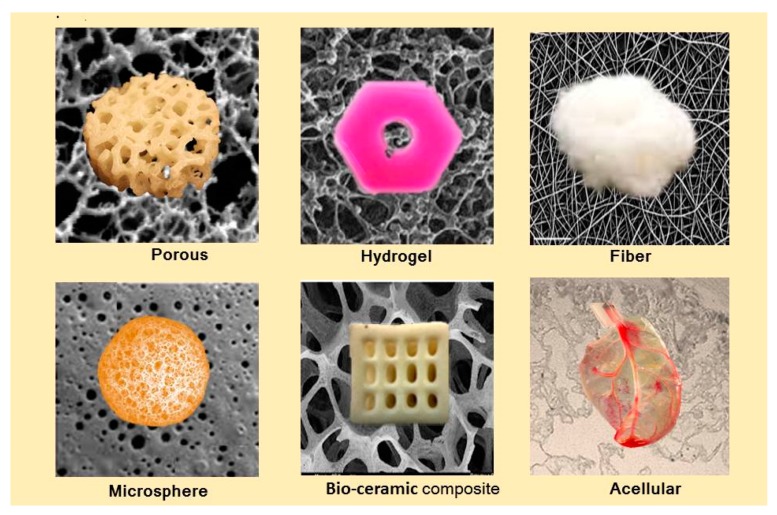
Different forms and morphological structures of scaffolds for biomedical applications.

**Figure 4 biomolecules-09-00619-f004:**
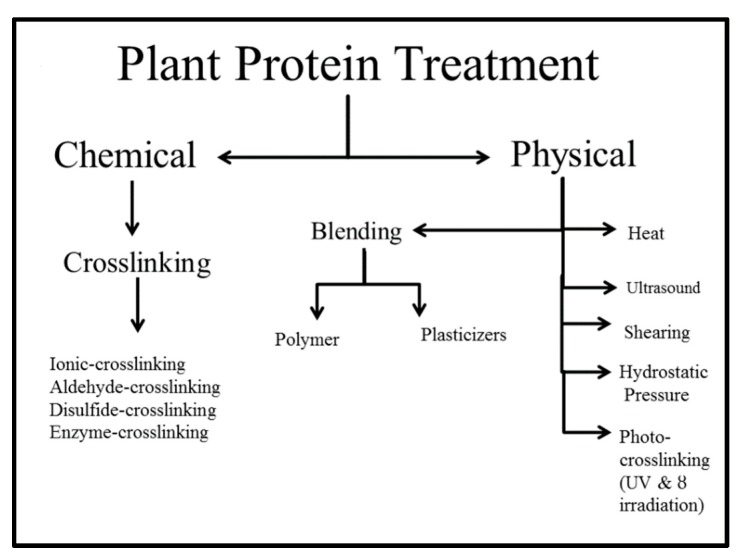
A schematic of different plant protein treatments.

**Figure 5 biomolecules-09-00619-f005:**
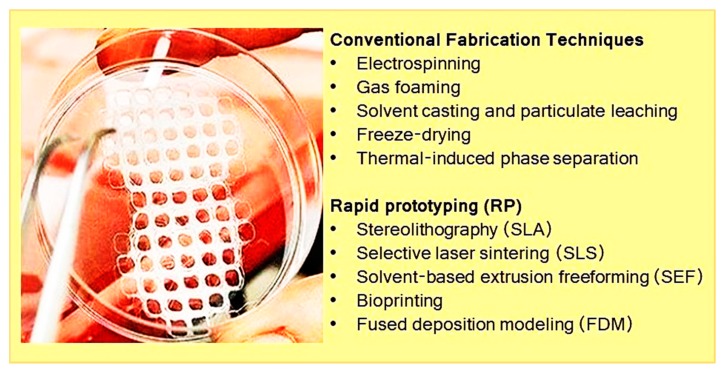
Different fabrication methods for producing scaffolds.

**Table 1 biomolecules-09-00619-t001:** A Comparison between properties of plant proteins and animal proteins. Data from Reference [20].

Property											
Protein		Molecular Weight (kDa)	Isoelectric Point	Platform				Solubility			Wet Mechanical Properties
				Micro/Nano Particles	Micro/Nano Fibers	Micro/Nano Film	Hydrogel	Water	Ethanol	Organic Solvents	
Soy		25–120	4.5–4.8	✓	✓	✓	✓	×	×	×	Good
Zein		19–25	6	✓	✓	✓	×	×	✓	✓	Fair
Wheat gluten	Gliadin	25–55	6.5	✓	✓	✓	×	×	✓	✓	Fair
	Gluten	35–100	6	✓	✓	✓	×	×	×	×	Good
	Glutenin	32–130	6.8–7.0	✓	✓	✓	×	×	×	×	Good
Collagen		300	4.7	✓	✓	✓	×	✓	✓	✓	Poor
Silk		250–450	3.8–3.9	✓	✓	✓	×	×	×	✓	Excellent
**Ref**		[21,22,23]	[21,22,23]	[24,25,26]	[27,28,29]	[30,31]	[32]	[27,33]			[11,34,35,36]

**Table 2 biomolecules-09-00619-t002:** Advantages and disadvantages of different scaffolds used in tissue engineering. Adapted from Reference [38].

Scaffold Types	Advantages	Disadvantages
Porous	High porosity provides a proper environment for extracellular matrix (ECM) secretion and nutrient materials to cells. Pore sizes exact to cell types avoid clustering of the cells, therefore preventing necrotic center formation.	Homogenous distribution of the cells is confined by a porous nature. Diverse pore sizes are needed for the particular cell types and are, thus, time consuming.
Fibrous	A highly microporous structure is best appropriate for adhesion, proliferation, and differentiation of cells. Slight inflammatory reaction upon implantation.	Surface functionalization is needed to make the nanofibers of these scaffolds.
Hydrogel	Extremely biocompatible and controlled biodegradation degree.	Low mechanical strength owing to soft structures.
Microsphere	Easily produced with controlled physical features suitable for slow or rapid drug delivery. Provides greater cell attachment and migration characteristics.	Microsphere sintering approaches are sometimes not compatible to the cells and decreases cell viability.
Composite	Highly biodegradable and offer mechanical strength. Greater absorbability.	Acidic derivatives are generated upon degradation. Insignificant cell affinity. Require tedious efforts to develop composite scaffolds.
Acellular	Natural ECM is maintained and consequently normal anatomical features are retained. Slight inflammatory and immune response with greater mechanical strength.	Partial decellularization is needed to avoid immune reactions.

**Table 3 biomolecules-09-00619-t003:** The most recent cellular and acellular reported studies using various forms of scaffolds for tissue engineering (TE) applications.

Scaffold Structure	Method	Cells/Factors/Animal Model	Type	TE	Ref
Collagen and denatured collagen (DCol)	Solution casting-freezing-thawing	Rabbit chondrocytes seeding	3D porous	Cartilage	[54]
Collagen (Col)/carbon nanotube (CNT)/chitosan(CS)/hydroxyapatite (HAP)	Freezing and lyophilization	-	3D porous	Bone	[77]
Poly(lactic acid) (PLLA)/ polycaprolactone (PCL), and collagen type I	Freeze-drying	Adipose tissue-derived mesenchymal stem cells seeding	3D porous	Skin	[92]
Silk fibroin	Freezing	-	3D porous (sponge)	-	[93]
Decellularized extracellular matrix (dECM)/gelatin/chitosan	-	Bone marrow mesenchymal stem cell (BMSC) seeding	Porous	Meniscus	[94]
Collagen/dECM/silk fibroin (SF)	3D printing	Pre-osteoblast MC3T3-E1 cells	3D micro-nanoporous	Bone	[43]
Collagen	3D Cell-printing	MC3T3-E1	3D porous	-	[44]
Collagen type I/agarose with sodium alginate	3D printing	Primary chondrocytes	3D porous hydrogel	Cartilage	[45]
α-TCP/collagen	3D printing combined with a cell-printing	MC3T3-E1 cells	3D porous	Bone	[46]
Polycaprolactone/polyvinyl acetate (PCL/PVAc)/poly(lactic-co-glycolic acid)- one morphogenetic protein 2 (PLGA-BMP2)	Electrospinning and electrospraying	Osteogenic and osteoconductive markers (OCN and OPN)	3D porous core-shell nanofibers	Bone	[47]
Hydroxyapatite/gelatin-chitosan	Coaxial electrospinning	Human osteoblast like cell line (MG-63)	3D porous nanofibers	Bone	[48]
Polycaprolactone (PCL) nanofibres/poly (lactic-co-glycolicacid) (PLGA) particles	Electrospinning	-	3D porous nanofibers	Bone	[49]
Hydroxyapatite (HA), 5CuHA and 5MgHA	Sol-gel and physio-chemical mixing	-	3D porous fibers	Bone	[50]
Chitosan/Sodium β-glycerophosphate/Gelatin (Cs/GP/Gel)	Freeze-drying	P3 bone mesenchymal stem cells (BMSCs)	3D porous	Cartilage	[51]
Polyurethane (PU), rosemary (RM) oil, and copper sulphate (CuSO4)	Electrospinning	Fibroblast cells	Two-dimensional (2D) porous nanofiber	Bone	[52]
Chitosan (CS)/nano-hydroxyapatite (n-HAP)	Solution casting/Freeze-drying	MC3T3-E1 cells	Porous polymer/bioceramic	Bone	[78]
Alginate/gelatin/nano-hydroxyapatite (n-HAP)	-	MG63 cells	Hydrogel	Bone	[79]
Chitosan-gelatin (CS-Gel)/graphene oxide (GO) and Montmorillonite (MMT)	Freeze-drying	Human osteoblast cells	Porous	Bone	[53]
Gelatin, alginate, and poly (vinyl alcohol)/silver hydroxyapatite	Cryogelation	MC3T3-E1 preosteoblast cells	3D porous spongy	Bone	[39]
Chitosan/alginate/hydroxyapatite/nanocrystalline cellulose	Freeze-drying	Fibroblast cells	3D porous	Bone	[80]
Bacterial cellulose (BC)/magnetite (Fe_3_O_4_)/hydroxyapatite (HA) nanoparticles	Ultrasonic irradiation	Mouse fibroblast L929 cells and osteoblast (MC3T3-E1 cell line)	3D microporous	Bone	[81]
Nipple-areolar complex (NAC) tissue	Decellularization	Bone marrow-derived mesenchymal stem cells (BMSCs)	Acellular	NAC	[85]
Decellularized myocardium extracellular matrix (ECM) and chitosan (CS)	Frozen and lyophilized	Human cardiac progenitor cells (hCPCs)	3D macroporous cardiac	Myocardial	[88]
Decellularized pig oesophagi	Decellularization	Human aortic smooth muscle cells (hASMCs) or human adipose-derived stem cells (hASCs)	Esophageal acellular	Esophageal muscle layers	[89]
Acellular spinal	Decellularization	Rat bone marrow mesenchymal stem cells/Neurotrophic factor 3 (NT-3)	Acellular spinal	Spinal cord	[90]
Human dura mater	Acellularized	-	Acellular dura mater	-	[91]
Poly(ethylene glycol) (PEG/poloxamer) (P407)	Photo-polymerization	Wistar rat thigh	Hydrogel	Artificial cornea periphery	[62]
Poly(ethyleneglycol)-poly(N-isopropylacrylamide) (PEGPNIPAAm), /poly(e-caprolactone) (PCL)	Free-radical polymerization	Human mesenchymal stem cells (hMSCs)	Hydride hydrogel	Cartilage	[63]
1% collagen microspheres and 0.3% collagen bulk	-	Human umbilical vein endothelial cells (HUVECs)/Eight-week old male C57/BL6 mice	Microspheres hydrogel	Dermal	[64]
Methacrylathed pullulan	Multiscale light assisted 3D printing	Epithelial and mesenchymal cells	2D and 3D hydrogel	-	[65]
Silk fibroin (SF)/gelatin/bacterial cellulose nanofibers (BCNFs)	3D printing and lyophilization	L929 cells	Hydrogel	-	[66]
Chitosan/silk (particles, micro and nanofibers)	3D printing	Human fibroblasts	Hydrogel	soft tissue	[67]

**Table 4 biomolecules-09-00619-t004:** Comparison of mechanical properties of soy protein fibers with other non-crosslinked plant protein fibers.

Fiber	Strength, MPa	Elongation, %	Modulus, GPa	Ref.
Soy protein	145 ± 10	8 ± 2	6.5 ± 1.7	[167]
Zein	36 ± 60	1.8 ± 5.0	-	[30]
Wheat gluten	115 ± 7	23 ± 2.7	5 ± 0.2	[168]
Gliadin	120 ± 10	25 ± 3.2	4.2 ± 0.4	[168]

**Table 5 biomolecules-09-00619-t005:** Mechanical properties of soy protein isolate (SPI) fibrous scaffolds for skin tissue regeneration.

SPI Fibrous Scaffolds	Method	Tensile Strength (MPa)	Young’s Modulus	Elongation (%)	Ref.
SPI (7 wt.%)/PEO(3wt.%)	ES	0.06 ± 0.01	110 ± 6 KPa	-	[164]
SPI (12 wt.%)/PEO(3wt.%)	ES	0.17 ± 0.006	171 ± 21 KPa	-	[164]
Hydrated SPI (5, 6, 7, 8%)/PEO (0.05%)	ES	0.1	-	-	[161]
SPI (10 wt.%)/PEO(4 wt.%) (40:60)	ES	2.3	-	9	[188]
Soy protein fiber	WS	145 ± 10	6.5 ± 1.7 GPa	8 ± 2	[166]

NOTE. ES: Electrospinning and WS: Wet spinning.

**Table 6 biomolecules-09-00619-t006:** Studies using soy protein-based matrices/scaffolds for tissue engineering (TE) applications.

Scaffold Structure	TE Application	Method	Encapsulated/Seeded Cell Type (Source)	Animal Model	Ref
SPI/micron-sized 45S5 bioactive glass (BG)	-	Electrospinning	Mouse embryonic fibroblast (MEF) cells	-	[186]
Tetracycline-loaded alginate/soy protein isolate (TCH-Alg/SPI)/polycaprolactone (PCL)	-	Co-axial electrospinning	Human dermal fibroblasts	-	[187]
Soy protein modified bacterial cellulose (BC)	bone	Electrospinning, ultrasound-induced self-assembly	MG-63 cells	-	[190]
Hydroxypropyl chitosan (HPCS)/soy protein isolate (SPI)	Skin	Crosslinking, solution casting, and evaporation	L929 cells	Rat	[191]
Ethylene glycol diglycidyl ether (EGDE)-crosslinked hydroxyethyl cellulose (HEC)/soy protein isolate (SPI)	Skin	Blending, crosslinking and freeze-drying	L929 cells	-	[192]
Soy protein isolate/bioactive glass	Skin	Solvent-casting	Mouse embryonic fibroblast (MEF) cells	-	[193]

**Table 7 biomolecules-09-00619-t007:** Studies using zein protein-based matrices/scaffolds for tissue engineering (TE) applications.

Scaffold Structure	TE Application	Method	Encapsulated/Seeded Cell Type (Source)	Animal Model	Reference
PCL/zein/gum arabic (GA)	Skin	Electrospinning	Fibroblast L929 cell	-	[214]
PCL/zein/GA/Calendula officinalis	Skin	Suspension, multilayer and two-nozzle electrospinning	Fibroblast L929 cell	-	[215]
Zein/(PSS-modified HAP) nanoparticles	Bone	-	MG-63 cell	-	[216]
HAP/zein	Bone	Solvothermal	Mouse bone marrow mesenchymal stem cells (MSCs)	-	[217]
Zein/chitosan/nanohydroxyapatite (nHAP)	Bone	Freeze-drying	MG-63 cell	-	[218]
PCL/zein coated 45S5 bioactive glass	Bone	Foam replication	-	-	[219]
Poly(ε-caprolactone)-thermoplastic zein/hydroxyapatite particles	Bone	scCO_2_ foaming	Osteoblast-like MG63 and hMSCs	-	[220]
Zein/calcium phosphate	Bone	Electrospinning/biomimetic mineralization process	Adipose-derived stem cells (ASCs)		[133]
rhBMP-2-loaded silica/HACC/zein	Bone	Solvent casting/Salt-leaching	hMSCs	-	[221]
Zein films	-	Solvent casting	Human liver cells (HL-7702) and mice fibroblast cells (NIH3T3)	-	[222]
CdS /zein	-	Electrospinning	MSCs and fibroblasts	-	[223]
Zein	Bone	Salt-leaching	MSCs	Rabbit	[224]
Zein/oleic acid Zein/citric acid	Bone	Salt-leaching porogen (Mannitol)	MSCs	Rabbit	[197]
PLGA/HAP/zein	Cartilage	Electrospinning	hUC-MSCs	-	[225]
Zein	Bone	-	HUVECs and MSCs	Rabbit	[226]
Zein polydopamine/bone morphogenic protein-2 (BMP-2) peptide conjugated TiO_2_	Bone	Electrospinning	Human fetal osteoblast	-	[227]
Zein/45S5 bioactive glass	Bone	Salt leaching	-	-	[228]
Zein/PLLA	Bone	Electrospinning	MSCs	-	[229]
Zein/gelatin	-	Electrospinning	Human periodontal ligament stem cells	-	[230]
Zein/gelatin	-	Force-spinning	Human fibroblasts	-	[231]
Poly(ε-caprolactone) (PCL)/zein	-	Electrohydrody- namic printing	Mice embryonic fibroblast (NIH/3T3) and human non small lung cancer cell (H1299)	-	[232]
Zein/silver-doped bioactive glass	Bone	-	MG-63 cells	-	[233]

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
