# Peer review of "Status of Plant Protein-Based Green Scaffolds for Regenerative Medicine Applications"

_biomolecules, 2019, doi:10.3390/biom9100619_

Round 1

Reviewer 1 Report

nothing

Author Response

The manuscript was revised for English.

Reviewer 2 Report

The current manuscript is a resubmission of the authors’ review on the status of plant protein-based green scaffolds for regenerative medicine applications “biomolecules-569248”. In the response letter, the authors tried to add some additional information to improve the novelty. Although they stated that this paper is different from the others and presents novel knowledge/insight, this review paper still highlights green (or environmentally friendly) materials and manufacturing processes for creating green tissue engineering materials. Unfortunately, in my opinion, this research group has actually reported a series of publications that emphasize the development of green tissue engineering materials. Here, the new sections include (I) plant-proteins-based nanocomposites, (II) plant protein-based electrospun nanofibers and films/natural extracts and (III) Aloe vera scaffolds. However, after a deep analysis of these parts, this reviewer still believe that the value of the current review manuscript is limited due to insufficient novelty. Actually, as mentioned in my comments during previous round of peer-review, the authors have published several review papers that deal with the green technology for processing biomaterials in tissue engineering. For example: Jahangirian H, Lemraski EG, Rafiee-Moghaddam R, Webster TJ. A review of using green chemistry methods for biomaterials in tissue engineering. Int J Nanomedicine. 2018;13:5953-5969. & Kalantari K, Afifi AM, Jahangirian H, Webster TJ. Biomedical applications of chitosan electrospun nanofibers as a green polymer - Review. Carbohydr Polym. 2019;207:588-600. Furthermore, they also focused on the fabrication of nanofibrous scaffolds for biomedical applications (please refer to the following paper: Stocco TD, Bassous NJ, Zhao S, Granato AEC, Webster TJ, Lobo AO. Nanofibrous scaffolds for biomedical applications. Nanoscale. 2018;10:12228-12255.). These earlier publications mainly focus on the report of part (II) concept. Furthermore, as reported in the literature, a review on the plant protein-based biomedical nanocomposites (i.e., part (I)) has given the readers similar scientific views (please refer to the following paper: Tian H, Guo G, Fu X, Yao Y, Yuan L, Xiang A. Fabrication, properties and applications of soy-protein-based materials: A review. Int J Biol Macromol. 2018;120:475-490.). In particular, Aloe vera for tissue engineering applications (i.e., part (III)) has also been reviewed by others before (please refer to the following paper: Rahman S, Carter P, Bhattarai N. Aloe Vera for Tissue Engineering Applications. J Funct Biomater. 2017;8:6.). I must regretfully say that since the biomolecules science/processing and tissue engineering/regenerative medicine community is not getting any substantial new messages from this article, I still unable to to recommend that the manuscript is suitable for consideration in publication to a high-quality journal "Biomolecules".

Author Response

We previously addressed this comment and added significantly new innovative sections to make the review novel.

Reviewer 3 Report

The manuscript represents a significant contribution to the science of Plant Protein-Based Scaffolds.

The manuscript should be supplemented with considerations on the physico-chemical properties of scaffolds important for their potential application. Try to consider the importance of porosity, size and shape of pores, surface charge, surface hydrophilicity or wettability, etc.

Author Response

We added a section as suggested emphasizing the role of porosity and other properties important for regenerative medicine.

Round 2

Reviewer 2 Report

The revised manuscript does not really give any professional response to this reviewer’s comments. From scientific point of view and personal peer-review standard, this reviewer still cannot support the publication of the authors’ current work due to insufficient academic novelty and scientific progress as compared to the authors’ own and others’ review-type articles. Although they stated that this paper is different from the others and presents novel knowledge/insight, this review paper still essentially highlights green (or environmentally friendly) materials and manufacturing processes for creating green tissue engineering materials. I must regretfully say that since the biomolecules science/processing and tissue engineering/regenerative medicine community is not getting any substantial new messages from this article, I still unable to recommend that the manuscript is suitable for consideration in publication to a high-quality journal "Biomolecules".

This manuscript is a resubmission of an earlier submission. The following is a list of the peer review reports and author responses from that submission.

Round 1

Reviewer 1 Report

The paper has very old references, you should add recent references.

Author Response

We have updated the references throughout the manuscript to be more recent.

Reviewer 2 Report

This study aims to review the status of plant protein-based green scaffolds for regenerative medicine applications. Actually, the authors have recently reported several review papers that deal with the green technology for the development of biomaterials in tissue engineering. For example: Jahangirian H, Lemraski EG, Rafiee-Moghaddam R, Webster TJ. A review of using green chemistry methods for biomaterials in tissue engineering. Int J Nanomedicine. 2018;13:5953-5969. & Kalantari K, Afifi AM, Jahangirian H, Webster TJ. Biomedical applications of chitosan electrospun nanofibers as a green polymer - Review. Carbohydr Polym. 2019;207:588-600. Furthermore, they also focused on the fabrication of nanofibrous scaffolds for biomedical applications (please refer to the following paper: Stocco TD, Bassous NJ, Zhao S, Granato AEC, Webster TJ, Lobo AO. Nanofibrous scaffolds for biomedical applications. Nanoscale. 2018;10:12228-12255.). Here, they pay particular attention to the “plant protein-based green scaffolds”. However, the potential of plant proteins such as zein, wheat gluten, and soy protein for fabricating biomaterials (i.e., nanofibers and nanoparticles) to tissue engineering applications has been carefully reviewed by others (please refer to the following paper: Reddy N, Yang Y. Potential of plant proteins for medical applications. Trends Biotechnol. 2011;29:490-498.). As compared to this previous review paper, the current manuscript seems to be limited by its insufficient academic novelty and scientific progress. In my opinion, it is highly desirable for the audiences to learn the new viewpoints from the review type-article. Unfortunately, the biomolecules science/processing and tissue engineering/regenerative medicine community is not getting any substantial new messages from this article. Regretfully, based on the above concern, I am not able to recommend that the manuscript is suitable for consideration in publication to a high-quality journal "Biomolecules".

Author Response

We thank the reviewer for his/her comments on our manuscript. It has been significantly revised as suggested, adding more recent studies, highlighting the novelty of this review, and even including new tables. Revisions are added in blue.

Importantly, we want to emphasize how this paper is different from the others that the reviewer mentions. The novelty of this review paper is that it highlights green (or environmentally friendly) materials and manufacturing processes for creating green tissue engineering materials. Green technologies are often overlooked in regenerative medicine as toxic solvents and synthetic materials harmful to our environment are commonplace in the literature today. We do not see this same emphasis in other papers published and firmly believe that this paper represents a new summary of green technologies to the regenerative medicine field.

Round 2

Reviewer 1 Report

All the notes have been reworked.

Reviewer 2 Report

The authors' revision is highly appreciated. Although they stated that this paper is different from the others. The novelty of this review paper is that it highlights green (or environmentally friendly) materials and manufacturing processes for creating green tissue engineering materials. However, this research group has actually reported a series of publications that emphasize the development of green tissue engineering materials. This reviewer has carefully examined the authors' several review papers that deal with the green technology for processing biomaterials in tissue engineering. For example: Jahangirian H, Lemraski EG, Rafiee-Moghaddam R, Webster TJ. A review of using green chemistry methods for biomaterials in tissue engineering. Int J Nanomedicine. 2018;13:5953-5969. & Kalantari K, Afifi AM, Jahangirian H, Webster TJ. Biomedical applications of chitosan electrospun nanofibers as a green polymer - Review. Carbohydr Polym. 2019;207:588-600. Furthermore, they also focused on the fabrication of nanofibrous scaffolds for biomedical applications (please refer to the following paper: Stocco TD, Bassous NJ, Zhao S, Granato AEC, Webster TJ, Lobo AO. Nanofibrous scaffolds for biomedical applications. Nanoscale. 2018;10:12228-12255.). Here, they pay particular attention to the “plant protein-based green scaffolds”. However, the potential of plant proteins such as zein, wheat gluten, and soy protein for fabricating biomaterials (i.e., nanofibers and nanoparticles) to tissue engineering applications has been carefully reviewed by others (please refer to the following paper: Reddy N, Yang Y. Potential of plant proteins for medical applications. Trends Biotechnol. 2011;29:490-498.). As compared to this previous review paper, the current manuscript seems to be limited by its insufficient academic novelty and scientific progress. According to my experiences on serving an academic editor and reviewer for several international journals (over 700 manuscripts including Advanced Functional Materials, Advanced Science, ACS Nano, and Biomaterials, and so on), this review manuscript indeed contains drawbacks regarding scientific progress in new insights into the "green tissue engineering polymers". Since the biomolecules science/processing and tissue engineering/regenerative medicine community is not getting any substantial new messages from this article, I still unable to to recommend that the manuscript is suitable for consideration in publication to a high-quality journal "Biomolecules".